# BtuB TonB-dependent transporters and BtuG surface lipoproteins form stable complexes for vitamin B₁₂ uptake in gut *Bacteroides*

Javier Abellon-Ruiz [1,3], Kalyanashis Jana [2,3], Augustinas Silale [1], Andrew M. Frey [1], Arnaud Baslé[1], Matthias Trost[1], Ulrich Kleinekathöfer [2] ✉ & Bert van den Berg [1] ✉

Vitamin B₁₂ (cobalamin) is required for most human gut microbes, many of which are dependent on scavenging to obtain this vitamin. Since bacterial densities in the gut are extremely high, competition for this keystone micronutrient is severe. Contrasting with Enterobacteria, members of the dominant genus *Bacteroides* often encode several BtuB vitamin B₁₂ outer membrane transporters together with a conserved array of surface-exposed B₁₂-binding lipoproteins. Here we show that the BtuB transporters from *Bacteroides thetaiotaomicron* form stable, pedal bin-like complexes with surface-exposed BtuG lipoprotein lids, which bind B₁₂ with high affinities. Closing of the BtuG lid following B₁₂ capture causes destabilisation of the bound B₁₂ by a conserved BtuB extracellular loop, causing translocation of the vitamin to BtuB and subsequent transport. We propose that TonB-dependent, lipoprotein-assisted small molecule uptake is a general feature of *Bacteroides* spp. that is important for the success of this genus in colonising the human gut.

Vitamin B₁₂ (cobalamin) is a complex organometallic cofactor and the most complex vitamin[1], consisting of a corrin ring containing a cobalt atom in the centre, coordinated with an upper ligand (such as adenosyl or methyl group) and a lower ligand anchored to the ring through a nucleotide loop (Fig. 1a)[2]. The upper ligand contains the chemical reactivity, directly participating in reactions, while the lower ligand provides functional specificity[3–5]. There are three different families of lower ligands; benzimidazoles, purines and phenolics[5]. The nature of this ligand is important as enzymes belonging to different species can require different lower ligands to be active[3,4,6]. Vitamin B₁₂ is involved in a wide variety of metabolic processes, and for many organisms it is an essential cofactor for the final enzymatic reaction of the L-methionine biosynthesis pathway in the cytoplasm[7,8]. Despite its many roles in eukaryotic and prokaryotic cells, only a small group of

microorganisms is able to produce vitamin B₁₂. However, its synthesis is energetically expensive, requiring ~30 enzymatic steps[9]. As a result, microorganisms have developed mechanisms to take up exogenous cobalamins, named the salvage route. In Gram-negative bacteria, the translocation of cobalamins presents challenges given that three different compartments need to be crossed: the outer membrane (OM), the periplasm and the inner membrane (IM). The best-characterised B₁₂ transport system in Gram-negatives is that of *Escherichia coli*, comprising the OM TonB-dependent transporter (TBDT) BtuB, the periplasmic binding protein BtuF, and the BtuCD ABC transporter located in the IM[10–12].

The human gut microbiome is the highly complex community of microorganisms in the gastrointestinal tract and has been implicated in many aspects of human health[13,14]. The vast majority of human

[1]Biosciences Institute, Faculty of Medical Sciences, Newcastle University, Newcastle upon Tyne NE2 4HH, UK. [2]School of Science, Constructor University, Campus Ring 1, 28759 Bremen, Germany. [3]These authors contributed equally: Javier Abellon-Ruiz, Kalyanashis Jana. ✉e-mail: ukleinekathoefer@constructor.university; bert.van-den-berg@newcastle.ac.uk

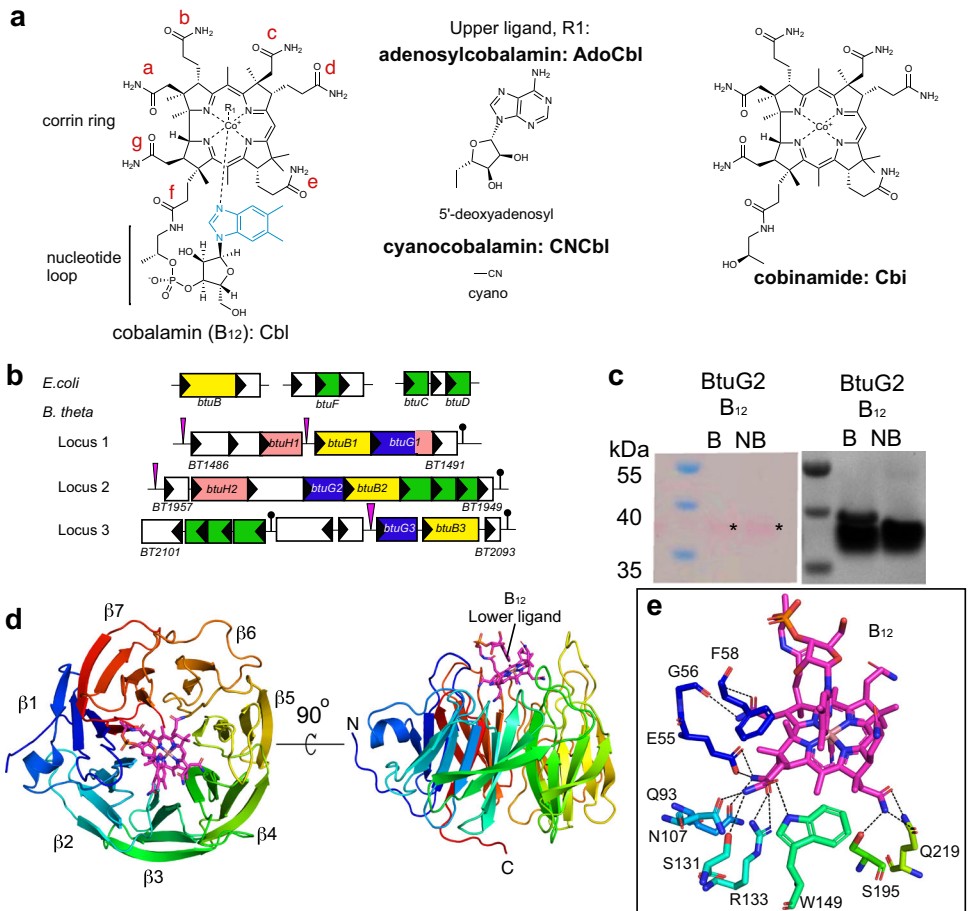

**Fig. 1 | BtuG2 has a beta propeller fold which binds corrinoids. a** Diagrams showing the structures of the corrinoids used for crystallography. Lower ligand is depicted in light blue in the "base-on" conformation. Side chains of the corrin ring are labelled with letters in red. **b** Genetic organisation of the $B_{12}$ transport system in *E. coli* (*btuBFCD*) and the three homologous loci in *B. theta*, showing the locations of BtuB (yellow), BtuG (blue) and BtuH proteins (pink), notice that BtuG1 has a BtuH domain fused. The pink triangles represent $B_{12}$-dependent riboswitches and the black lollipops transcription terminators. The inner membrane ABC transporters are in green. **c** SDS gel showing faint pink bands, indicated with an asterisk on the left panel, corresponding to CNCbl. The boiled sample also shows a minor lower mobility band (-15% of the sample) which corresponds to the fraction of BtuG2 that has lost CNCbl after boiling. The right panel shows the same gel after Coomassie staining (B, boiled; NB, non-boiled). Gels are representative of three independent replicates, uncropped gels in Source Data. **d** Cartoon representation of BtuG2 (in rainbow colour, N terminus in blue) bound to CNCbl (magenta). Note that the lower ligand is pointing outwards (right panel). **e** Close-up of the residues forming hydrogen bonds (black dashed lines) with CNCbl.

commensal microbes are located in the distal gut with bacterial densities of -10^{12}/g luminal content[15], making competition for resources likely severe. To understand the requirements for being competitive within the gut it is important to understand how small-molecule acquisition occurs. Despite this, most of these uptake processes are poorly characterised, especially for non-Proteobacterial members of the gut microbiota.

The Gram-negative bacterium *Bacteroides thetaiotamicron* (*B. theta*), like the vast majority of the Bacteroidetes phylum, cannot make vitamin $B_{12}$ but possesses several $B_{12}$-dependent enzymes[16]. One of these enzymes is the methionine synthase MetH, making $B_{12}$ an essential nutrient for this bacterium. *B. theta* therefore serves as a model for the salvage routes of cobalamins by the most abundant Gram-negative phylum in the gut[17]. Interestingly, while model organisms like *E. coli* have just one vitamin $B_{12}$ uptake locus, *B. theta* has three, together containing 24 proteins (Fig. 1b), suggestive of the importance of $B_{12}$ uptake for this organism[18]. All three loci contain one copy each of the TonB dependent transporter (TBDT) BtuB (BtuB1-3) as well as one copy of BtuG, which is always located adjacent to BtuB on the genome[19]. The presence of multiple outer membrane $B_{12}$ transporters is a widespread feature in *Bacteroides* (and Bacteroidetes in general), with up to four BtuBs[18]. Competition assays have shown

that strains lacking *btuB2* or *btuG2* have fitness defects in vivo and in vitro, and are rapidly outcompeted by wild-type cells, suggesting that locus 2 might encode the primary uptake system in *B. theta*[18]. Interestingly, BtuG2 is a surface-exposed OM lipoprotein with an extremely high affinity for $B_{12}$ (sub-pM) and is able to remove the vitamin from human intrinsic factor (IF), which binds $B_{12}$ in the small intestine prior to absorption[19]. However, while BtuG2 is probably highly efficient in scavenging $B_{12}$ from the gut lumen, it is unclear how the vitamin is transferred to BtuB2 for uptake. Genomic studies show that all *Bacteroidetes* $B_{12}$ transport loci include homologues of BtuB and almost all encode BtuG, suggesting the two proteins work together to bring $B_{12}$ into the cell[18,19].

Here we present X-ray crystal structures of *B. theta* BtuG2 complexed with cyanocobalamin (the manufactured and most stable form of $B_{12}$, CNCbl), adenosylcobalamin (one of the biologically active forms of vitamin $B_{12}$ with the bulkiest upper ligand, AdoCbl) as well as the precursor cobinamide (Cbi), which lacks the lower ligand and the nucleotide loop. The positively charged cofactors (Co is charged +3e)[20] are bound at the same site on the negatively charged face of the BtuG2 beta-propeller and make many polar and hydrophobic interactions with protein residues. Molecular dynamics (MD) simulations reveal that BtuG2 can attract cyanocobalamin to its binding site over large

distances (>35 Å), establishing it as an efficient $B_{12}$ capturing device. BtuG2 forms a stable complex with BtuB2 in the OM, allowing co-purification and structure determination of BtuB2G2 by X-ray crystallography. The structure of the BtuB2G2 complex demonstrates that BtuG2 caps the transporter, reminiscent to recently described SusCD systems involved in glycan uptake[21]. Interestingly, however, the BtuB2G2 structure as well as the cryo-EM structure of the BtuBG complex from locus 1 (BtuB1G1) show closed transporters in the absence of substrate, which is different from SusCD systems. A cryo-EM structure of BtuB3G3 bound to CNCbl, MD simulations and functional data suggest a pedal bin uptake mechanism and provide an explanation for how $B_{12}$ is released from BtuG and then transferred to BtuB for subsequent uptake. Together with OM proteomics data, our study suggests that lipoprotein-assisted small molecule uptake operates for most, and perhaps all, TBDTs of Bacteroides spp, and we propose that this is one reason why these microbes are so successful within the human gut.

## Results

### BtuG2 binds a variety of corrinoids

To identify the $B_{12}$ binding site of BtuG2, we overexpressed the protein without its signal sequence and lipid anchor in *E. coli*. To load BtuG2 with the ligand before the size exclusion chromatography (SEC) purification step, we added CNCbl at a 1:2 molar ratio (protein:vitamin). The non-bound vitamin elutes later than the BtuB2-CNCbl complex, allowing us to eliminate excess vitamin in the sample. As observed previously, CNCbl binds to BtuG2 with very high affinity[19]. The BtuG2-CNCbl complex is extremely stable, given that a boiled sample run on a denaturing SDS-PAGE gel shows a visible pink band around 40 kDa, suggesting that some CNCbl is still bound to BtuG2 (Fig. 1c). An initial BtuG2-CNCbl structure was solved to 1.9 Å resolution using cobalt-based single-wavelength anomalous diffraction (Co-SAD), showing unambiguous electron density for CNCbl. This model was used to solve an additional dataset at slightly higher resolution (1.7 Å) (Supplementary Table 1). BtuG2 adopts a seven-bladed beta-propeller fold (Fig. 1). All blades have 4 beta strands except blade number 6 (β6), which has only two. The propeller blades define a central cavity that forms the $B_{12}$ binding pocket with an interface area of 698 Å² as measured via PISA[22]. CNCbl is oriented with the cyano group (upper ligand) pointing towards the protein centre while the lower ligand, present in its "base-on" conformation[20], is completely exposed to the exterior, with the nucleotide loop sitting in the cleft formed between blades 1 and 7 (Fig. 1 and Supplementary Fig. 1b). There are 13 hydrogen bonds between BtuG2 and CNCbl, all mediated by the amide groups of the substituents from side chains a, b, c and g of the corrin ring (Supplementary Fig. 1b, e) and involving protein residues from blades 1 to 4. CNCbl binding is further stabilised by Van der Waals forces provided by 11 residues, evenly distributed across the blades. The cavity that contains the cyano upper ligand is narrowed by Trp194, Trp272 and Tyr316 (numbering based on the annotated Uniprot sequence, including the signal peptide). To explore if this narrow pocket could accommodate a cobalamin with a bulky upper ligand, we obtained a structure to 2.3 Å resolution of BtuG2 co-crystallised with AdoCbl. In this structure the side chain of Trp272 has rotated, enlarging the cavity to fit the adenosyl group. The interaction is stabilised by stacking forces from Trp272 and a hydrogen bond between the hydroxyl group of Tyr335 and the nitrogen in position 3 of the pyrimidine ring of the adenosyl group (Supplementary Fig. 1c, Fig. 1d and Fig. 1e). For both structures the hydrogen bonds interacting with the side chains of the corrin ring are identical. A Consurf analysis[23] shows that while the degree of conservation of the protein surface is low overall, the residues involved in forming hydrogen bonds with the ligands are highly conserved (Supplementary Fig. 2). Thus, given that the lower ligand does not interact with the protein, the structures suggest that BtuG2 can bind many different, if not all, corrinoids. To investigate whether

the lack of lower ligand would affect the binding mode, we also determined the BtuG2-Cbi crystal structure, using data to 1.35 Å resolution. The orientation of the ligand is virtually the same as for the other structures (Supplementary Fig. 1e), confirming BtuG2 as a versatile corrinoid binder.

### Long-range electrostatic attraction of CNCbl by BtuG2

To study the BtuG2-CNCbl interaction in real time, unbiased MD simulations were performed using the crystallographic model for CNCbl-BtuG2. For analysis, we defined the $B_{12}$ binding pocket as those residues that have an atom within 3 Å distance of the ligand (Supplementary Fig. 3a). During the 1 μs-long simulation, the centre of mass (COM) distance between the ligand and the binding pocket varied less than 1 Å, indicating that CNCbl is bound very strongly and only wiggles slightly within the binding pocket (Supplementary Fig. 3b). The number of hydrogen bonds fluctuates around a constant value ($8.45 \pm 1.5$) and a superposition of the ligand before and after the simulation shows an almost identical orientation, supporting the notion that the interaction is highly stable (Fig. 2a). A calculation of the short-range electrostatic interactions between the ligand and the binding pocket results in large electrostatic interaction energies of around −30 kcal/mol throughout the simulations, explaining the stability of the ligand within the pocket.

In order to assess the CNCbl binding mechanism we analysed apo-BtuG2 using unbiased MD simulations. To this end, the CNCbl molecule was removed from the crystallographic model and the structure was simulated for 1 μs. The structure relaxed during the first half of the simulation and did not significantly change thereafter (Supplementary Fig. 3d). In the absence of the ligand, loops Asp236-Pro249 and Lys266-Ser274 (both from blade 5, henceforth named β5A and β5B loops) moved outwards away from the binding site by 17 and 7 Å, respectively (Fig. 2b). It is worth pointing out that loop β5 A contains Tyr239 which, in the crystal structures, contributes to CNCbl and AdoCbl binding through van der Waals (vdW) forces or via a hydrogen bond with Cbi. Additionally, loop β5B contains Trp272, providing vdW forces for CNCbl and Cbi binding. Moreover, the side chain of this residue rotates to allow AdoCbl binding (Supplementary Fig. 1c). No significant structural changes other than these two displaced loops and the C terminus were detected upon removal of CNCbl. Interestingly, the crystal structure of apo-BtuG2 (PDB 3DSM) is identical to that of BtuG2-CNCbl (Supplementary Information Fig. 1f), indicating that crystallisation of the apo protein induces a ligand bound-like conformation.

To explore the efficiency of CNCbl capture, we next ran three unbiased MD simulations of 4 μs in total, each starting with a CNCbl molecule placed at a different arbitrary initial position 8.5, 37.1 and 34.9 Å away from the binding pocket (Supplementary Fig. 4a). Strikingly, in all these simulations, regardless of the initial distance and orientation of the CNCbl molecule, the ligand moved rapidly into the binding site. The COM distance analysis shows that during the first μs, CNCbl binds via an association-dissociation process during which the ligand can move away from BtuG2 (sometimes almost doubling the furthest starting point distance and reaching the maximum distance possible in the present periodic simulation box) (Supplementary movie 1). Once the ligand is bound in the right conformation, it stays in the pocket for the remainder of the simulation (Fig. 2c). This strong attraction over long distances suggests a long-range electrostatic interaction. In all three simulations but after different times, CNCbl eventually adopts a similar conformation to the crystal structure (Supplementary Fig. 4d–f). BtuG2 has many negative charges (net charge −18$e$) leading to a strong electric field that attracts CNCbl to the binding pocket. A dense set of electric field lines points towards the binding pocket, explaining the attraction of an electrically neutral but strongly dipolar ligand with a dipole moment of 6.7D towards the binding pocket (Fig. 2e and

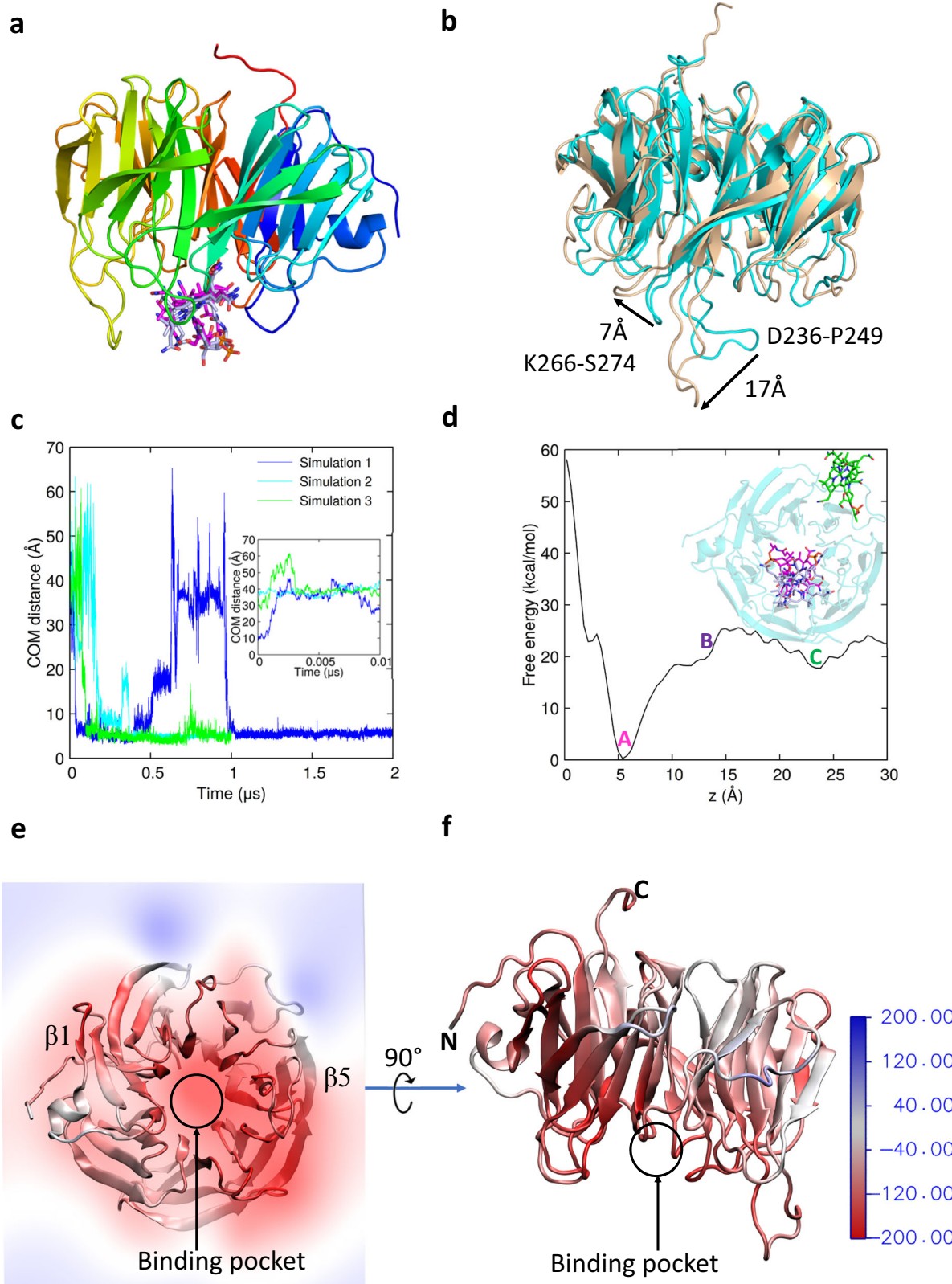

Supplementary Fig. 5a, b). Besides the electrostatic interactions, our MD studies show that conformational changes from loop β5A play a key role in ligand binding. Due to the displacement of this loop when relaxing the apo-BtuG structure, the binding pocket became much bigger than in the crystal structure. Upon binding, the ligand is held by loops β5A and Asn54-Thr64 like a pair of pliers (Supplementary Fig. 3e, f and Supplementary movie 1). Free energy calculations show

that the binding free energy for CNCbl is around $\Delta G = 20.0$ kcal/mol (Fig. 2f). This corresponds to a dissociation constant of $1.4 \times 10^{-14}$ M calculated using $k = e^{-\Delta G/RT}$ at 300 K, which is in reasonable agreement with the experimental value of $1.93 \times 10^{-13}$ M reported previously[19], considering the calculation approximations involved and the difficulties in measuring extreme affinities via surface plasmon resonance.

**Fig. 2 | BtuG2-CNCbl binding dynamics. a** Representative snapshot of the CNCbl-BtuG2 at the end of the 1-μs-long MD simulation performed using the BtuG2-CNCbl crystal structure. Included are the overlay of the final ligand position at the end of the unbiased (carbon atoms in blue) and the position in the crystal structure (carbon atoms in magenta). **b** Superposition of relaxed BtuG2 and the crystal structure. The displaced loops are annotated (cyan for the crystal structure, wheat for the relaxed structure). Arrows indicate loop movements away from the binding pocket during relaxation **c** COM distances between the binding pocket of the relaxed structure and the ligand initially placed arbitrarily at 8.5, 37.1 and 34.9 Å. The unbiased simulations 2 and 3 where stopped after 1 μs as CNCbl was stably bound. **d** Free energy profile for the binding of CNCbl to BtuG2 as a function of the

COM distance between the two binding partners. The free energy of binding, i.e., the difference between the lowest energy and the energy at large COM distances, is about 20 kcal/mol. Binding positions of the CNCbl on the free energy surface are shown in magenta (A, binding pocket), in light blue (B, 12–14 Å away from binding pocket), and in green (C, 25 Å away from binding pocket). **e, f** The electrostatic potential map of BtuG2 is predominantly negative because of the net charge of −18$e$, explaining why BtuG2 attracts the positively charged or zwitterionic CNCbl/AdoCbl/Cbi molecules. The BtuG2 conformation in this figure has been obtained by removing the CNCbl from the CNCbl-bound-BtuG2 crystal structure. The colour scale ranges from −200 for the negative surface (red), to 200 for the positive surface (blue) in units of $k_B T/e = 26$ mV at 300 K.

To better understand ligand binding, we calculated the short-range electrostatic interaction energy between the highly dipolar CNCbl and the negatively charged binding site of BtuG2 along the unbiased MD trajectories. Comparing the configurations and electrostatic energies suggests that binding occurs for values below −20 kcal/mol. Thus, once the electrostatic interaction energy between CNCbl and BtuG2 reaches values below −20 kcal/mol, CNCbl is strongly bound to the binding site of BtuG2 (Supplementary Fig. 4b), as supported by the number of hydrogen bonds formed during the simulations (Supplementary Fig. 4c).

### BtuB2 forms a stable complex with BtuG2

The presence of *btuB2* in an operon with *btuG2* suggests that the proteins could be working together in B$_{12}$ acquisition. To explore this possibility, we added a C-terminal hexa-histidine tag to chromosomal *btuB2* in *B. theta* and replaced the wild-type promoter with a strong constitutive promoter (Methods). Expression levels of BtuB2 in minimal media containing methionine were low (~0.05–0.1 mg/l culture), but sufficient for purification via IMAC and SEC. Analysis of the sample on a SDS page gel showed two clear bands upon boiling the sample (75 kDa and 40 kDa; Fig. 3a). The molecular weight of the upper band agrees with that of BtuB2 (77 kDa; Bt1953) and mass spectrometry analysis demonstrated that the ~40 kDa band corresponds to BtuG2 (Bt1954). Without boiling only one band is present, indicating that both proteins form a complex even in SDS.

The structure of the complex (BtuB2G2) was solved to 3.7 Å resolution by X-ray crystallography (Supplementary Table 1). The structure shows a typical TBDT fold for BtuB2 with an N-Terminal plug domain (pfam Domain PF07715) occluding a 22-stranded-β-barrel (pfam Domain PF00593). There is no electron density for the TonB box, suggesting that the N-terminal region is disordered despite the absence of ligand. BtuG2 is located on top of the barrel, forming an extracellular lid with a large buried interface area of 3463 Å (Fig. 3b). The BtuG2 model is complete, and shows the lipid anchor of the N-terminal cysteine (residue 32 in the full-length sequence) at the back of the complex (Fig. 3b, c). The linker between Cys32 and the beta propeller runs parallel to extracellular loop 2 (EL2; Ser202-Tyr239) from BtuB2 (Fig. 3b, d) and is constrained by the BtuB2 barrel via 4 hydrogen bonds provided by EL2 and EL11 (Fig. 3b and Supplementary Fig. 6; EL: extracellular loop). The BtuG2-BtuB2 interface is stabilised by ~40 hydrogen bonds that are evenly distributed across the interface, and 6 salt bridges. (Fig. 3 and Supplementary Fig. 6). The electrostatic interactions between the BtuG2 linker and EL2/EL11 offer a stable anchoring point which could be acting as a hinge to allow the opening of the lid. Gratifyingly, based on the BtuG2-CNCbl structure, the B$_{12}$ binding site of BtuG2 faces BtuB2. However, the BtuG2 lid caps BtuB2, and the BtuG2 binding site is solvent excluded and clearly inaccessible to any B$_{12}$. Moreover, despite the presence of 1.5 molar equivalents of CNCbl during crystallisation, no ligand is present in the crystallised complex. Indeed, BtuB2 loop EL8 is occupying the B$_{12}$ binding pocket region (Fig. 3e) and would clash with the ligand. Therefore, for BtuG2 to be able to bind B$_{12}$, a conformational change is required that opens the BtuG2 lid to expose the binding site. However,

it was not possible to load purified BtuB2G2 with CNCbl in vitro without destabilising the complex, suggesting that the closed structure is very stable in detergent. A ConSurf analysis[23] shows that the TBDT surface and periplasmic loops are poorly conserved, unlike the BtuB2 extracellular cavity which is highly conserved and contains the B$_{12}$ binding site. (Supplementary Figs. 2 and 14).

### The cryo-EM structure of BtuB1G1 also shows a closed complex

Previous work on SusCD complexes and the peptide transporter RagAB has shown that X-ray crystallography selects for closed states of dynamic transporters[21,24]. To obtain more experimental insight into the dynamics of BtuG lid opening, we determined the structure of the BtuB1G1 (i.e. the BtuBG complex of locus 1; Fig. 1b) complex by cryo-EM. BtuG1 is ~70 kDa in size due to the presence of a C-terminal extension of ~250 residues that is homologous to the BtuH2 protein that was recently shown to bind B$_{12}$ on the cell surface of *B. theta*[25]; thus, BtuG1 contains two B$_{12}$ binding sites. The additional BtuH domain increases the size of the extra-membrane part of the complex substantially, facilitating structure determination by cryo-EM. We obtained ~3.2 Å maps for the complex in the absence of CNCbl (Fig. 4a; Supplementary Table 2 and Supplementary Information Figs. 7 and 8). The additional BtuH domain (BtuG1 H-domain) is clearly visible and appears to be docked to BtuG1, in a position that would not allow B$_{12}$ binding (Fig. 4d). The resolution of the BtuG1 H-domain was substantially worse than the rest of the complex, implying some degree of conformational flexibility. Interestingly, and contrasting with substrate-free structures of SusCD glycan transporters[26], the BtuB1G1 EM structure is very similar to the BtuB2G2 crystal structure with a closed BtuG lid, and there are no particle populations with open BtuG1 lids (Fig. 4b). We also collected data for BtuB1G1 in the presence of 2 molar equivalents CNCbl, but analysis of the dataset suggested a destabilising effect on the DDM-solubilised complex caused by the presence of CNCbl, analogous to BtuB2G2 (Fig. 4c). However, this dataset was smaller compared to that in absence of the vitamin, and we cannot exclude that the low quality of the data was due to an insufficient number of particles or a poor-quality grid.

### Insights into BtuB2G2 B$_{12}$ lid opening from molecular dynamics

To quantify the opening of the lid, we selected the Cα atoms of Tyr37 from the BtuG2 linker and of Pro468 from BtuB2 EL7, whose relative positions remain relatively stable during the simulations, and of Asp236 from BtuG2 located on the opposite site of the hinge, i.e., in the widest opening region (Supplementary Information Fig. 9a-c). The spatial relation of the three Cα atoms (two static and one mobile) allows calculation of an aperture angle α. The angle is ~18° in the closed crystal structure (Supplementary Information Fig. 9d). Two unbiased simulations with different initial atomic velocities at $T = 300$ K gave mixed results. In one 2-μs-long simulation there is a short-lived opening of up to $α = 25°$, but in the other one the aperture remains similar to that in the crystal structure (Supplementary Information Fig. 9d). In the simulation where transient opening occurs, BtuG2 moves away from BtuB2 in a lid-like motion, with EL2 of BtuB2 and the BtuG2 linker acting as a hinge. To further explore opening, we ran a 5-μs-long

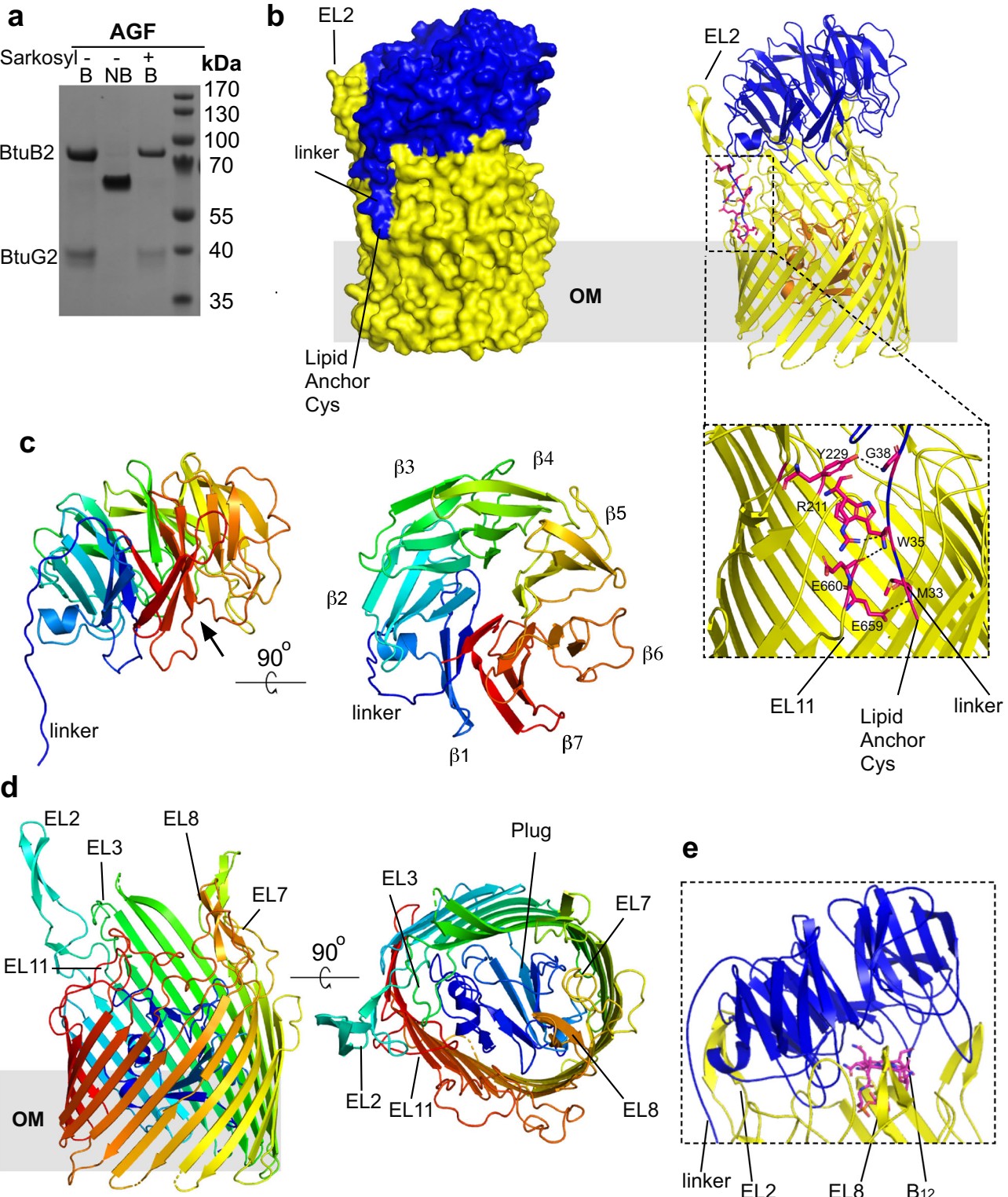

**Fig. 3 | BtuG2 and BtuB2 form a stable complex. a** Representative SDS-PAGE gel (n = 3 independent experiments) showing that BtuG2 co-purifies with BtuB2. B boiled samples, NB non-boiled sample. A sarkosyl pre-extraction step did not improve the purity of the sample and suggests that the ~40 kDa band is an OMP (sarkosyl selectively solubilises the cytoplasmic membrane, removing IM impurities[78]). Uncropped gels in Source Data. **b** Surface (left panel) and cartoon (right panel) views from the side for BtuB2G2 (BtuB2 β-barrel in yellow, plug in orange; BtuG2 in blue). EL denotes extracellular loop, β denotes β-propeller blade. The close-up view shows hydrogen bonds stabilising the linker (black dashed lines,

residues in red sticks). For clarity, loops are smoothed and the plug has been removed in this close-up view. **c** Lateral view of BtuG2 in rainbow colouring (orientation as in **b**; N-terminus in blue) and rotated by 90° to show the exterior surface. Black arrow indicates the B$_{12}$ binding site. **d** Lateral view of BtuB2, and rotated 90° showing the plugged pore. **e** Close-up view of the cartoon representation of BtuG2-BtuB2 (smooth loops), showing the BtuG2 CNCbl binding site in the complex. View was obtained after superimposing the BtuG2-CNCbl crystal structure onto BtuG2 in the complex). Note how EL8 from BtuB2 would clash with CNCbl.

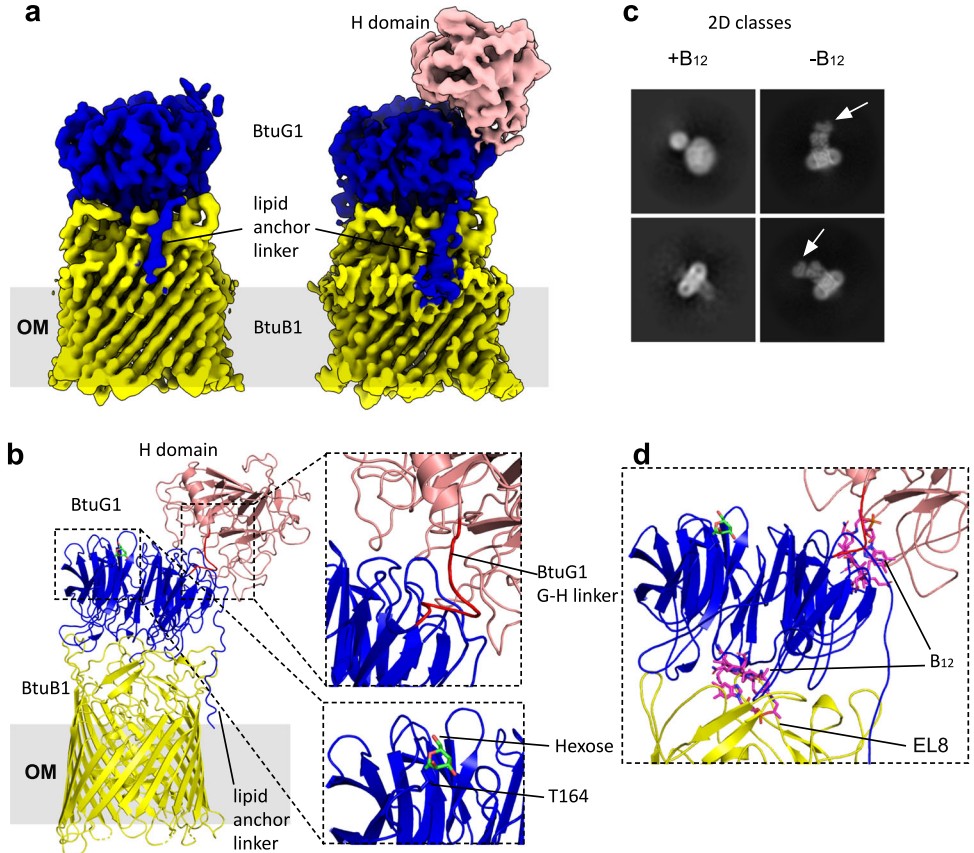

**Fig. 4 | The cryo-EM structure of BtuB1G1 is closed. a** Cryo-EM maps used to build BtuB1 and BtuG1 G domain (left panel) and BtuG1 H domain (right panel). BtuB1 in yellow, BtuG1 G-domain in blue and BtuG1 H-domain in salmon. **b** Cartoon model showing BtuG1 and BtuG1, colours as in **a**. The linker between the domains G and H is shown in red for spatial reference; note it has been modelled into very poor density (top close up view). The bottom close-up view shows an O-Glycosylation site[79] for BtuG1 at residue Thr164. **c** Representative 2D classification images from samples with and without B$_{12}$. White arrows denote the BtuG1 H-domain. **d** Close-up view of a cartoon model for BtuB1G1 showing the two B$_{12}$ binding sites occupied. B$_{12}$ has been placed by superimposing BtuH2-B$_{12}$ (7BIZ)[25] and BtuG2-B$_{12}$ onto BtuB1G1. Note the steric clashes of the modelled B$_{12}$ molecules with BtuB EL8 and with the BtuG1 G-domain.

simulation (Supplementary Information 9e, f). This revealed fluctuations up to $\alpha = 28°$, but no stable lid opening. To assess whether the large number of hydrogen bonds (~40) and salt bridges between BtuB2 and BtuG2 were hindering lid opening, we also ran two simulations at 400 K. In these simulations, we observe short-lived, partial lid openings of up to $\alpha = 37°$, suggesting that lid opening requires energy input and/or longer time scales (Supplementary Information 9g). A free energy simulation shows that the energy associated with the opening of the lid has a minimum for $\alpha = 18–22°$ (i.e., the closed state) but quickly increases to values of over 40 kcal/mol for $\alpha > 40°$ (Fig. 5a). These energies might be slightly overestimated due to shortcomings of classical MD simulations[27], but it is clear that opening angles of $\alpha > 40°$ are highly unlikely at ambient temperatures on microsecond timescales in our in silico setup. Despite this, an analysis of the electrostatics of the partially open BtuB2G2 complex suggests that BtuG2 will still attract CNCbl to its binding site (Supplementary Fig. 10).

One complication with the opening angle analysis is that any movement of the reference atoms relative to each other will affect $\alpha$, not just those caused by bona fide lid opening. For this reason, and to allow comparison with SusCD-like systems, we also measured the angles between the plane of the OM and the surface of the back of the SusG lid, which is stable during the simulations (Fig. 5b, c). Analysed this way, it is clear that the lid opening for BtuB2G2 is very modest even at 400 K (at most ~7°), and the apparent opening suggested by the structures is mostly caused by the outward movement of loop EL5 at the front of the complex (Fig. 5c). The small openings achieved in BtuB2G2 via MD simulations contrast with the stable open states

observed via cryo-EM for *B. theta* Bt1762-63 (SusCD)[26] and *Porphyromonas gingivalis* RagAB[24], which have openings of ~26° and ~52° (Fig. 5d).

It should be noted that, for the sake of simplicity and because the composition of the *B. theta* OM is not well-established (Methods), we used an OM composed of phospholipid in our simulations. We speculate that the presence of highly negatively charged lipo-oligosaccharide (LOS)[28] in the OM could conceivably lower the stability of the closed state of the transporter, e.g. via interactions with BtuB ELs.

## The BtuB3G3 cryo-EM structure with bound CNCbl provides insights into substrate release

To test whether the BtuBG complex from locus 3 (Fig. 1b) would be more amenable to substrate loading we replaced its native promoter with the strong constitutive promoter (P1E6) and purified 4 l of culture (methods). Yields of ~0.4 mg of BtuB3G3 per litre were obtained, which is 5–10-fold higher than for both other BtuBG complexes. This allowed assessment of the effect of detergent choice on complex stability in the presence of CNCbl. Indeed, in the presence of CNCbl the BtuB3G3 complex purified in LDAO is unstable (Supplementary Fig. 11a). However, BtuB3G3 incubated with excess CNCbl in the mild detergent Lauryl Maltose Neopentyl Glycol (LMNG) eluted as a single peak and had a pink colour after SEC, suggesting the presence of bound CNCbl. We determined the structure of this complex by cryo-EM (Supplementary Fig. 12). Data analysis produced maps of sufficient quality to build two slightly

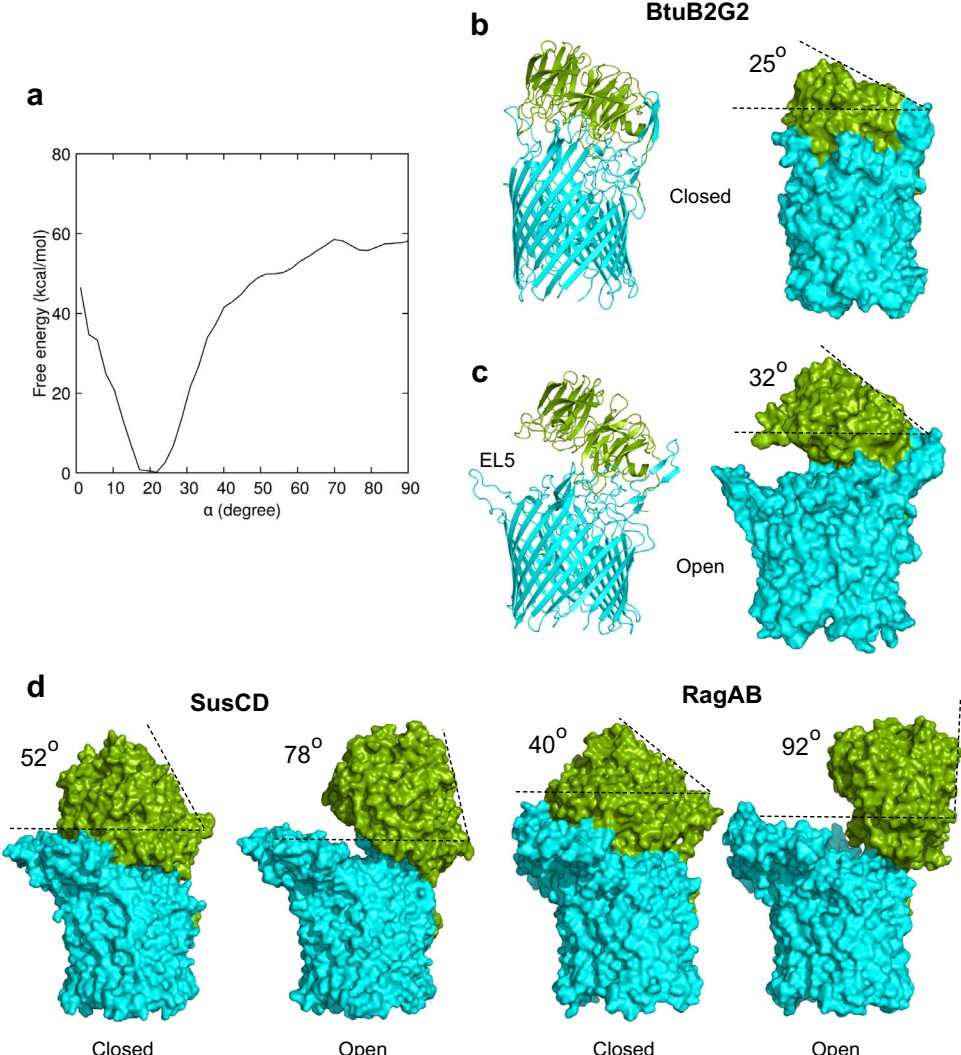

**Fig. 5 | BtuG2 lid opening investigated by molecular dynamics. a** Free energy profile of BtuG2 lid opening as a function of the angle *α*. See text and Supplementary Fig. 9 for details. **b**, **c** Surface representations of the closed state of BtuB2G2 (**b**) and the maximum open state observed during simulations at 400 K (**c**). The opening angles listed are defined by the dashed lines. BtuG2 is in green and BtuB2 in cyan. **d** Surface representations of the closed and open states of *B. theta* Bt1762-63 (PDB ID 6ZM1) and *P. gingivalis* RagAB (PDB ID 6SMQ).

different models, named state 1 and state 2 (Supplementary Figs. 8 and 13). Both states show density for a CNCbl molecule bound to BtuG3 (Fig. 6a and Supplementary Fig. 11b). To compare CNCbl binding to BtuB3G3 and BtuG3 alone, we also determined the crystal structure of BtuB3-CNCbl to a resolution of 2.6 Å. A superposition of BtuG3-CNCbl onto BtuB3G3-CNCbl (state 1) has a Cα RMSD of -0.6 Å (Fig. 6b), demonstrating that CNCbl binding to BtuG3 and BtuB3G3 is virtually identical. Moreover, BtuG2 and BtuG3 share 80% sequence identity, and a superposition of both structures bound to CNCbl shows that residues binding the ligand are the same and the overall structure is identical (Cα RMSD - 0.4 Å; Supplementary Fig. 11c–e). Thus, we conclude that CNCbl binding by BtuG2 and BtuG3 is the same. Since BtuB2 and BtuB3 are also very similar (74% sequence identity), a comparison of the apo BtuB2G2 structure with the CNCbl-bound BtuB3G3 structure could provide insights into the crucial question as to how a tightly BtuG-bound molecule of CNCbl is transferred to BtuB. In apo BtuB2G2, the tip of EL8 of BtuB2 is occupying the CNCbl binding site of BtuG2 (Fig. 6e). By contrast, in BtuB3G3-CNCbl (state 1), the EL8 loop has moved outwards to allow the vitamin to bind. In state 2, while density for the tip of EL8 is absent, the visible part at the base (Fig. 6e; green) occupies an intermediate position between that in the BtuB2G2 apo structure

(Fig. 6e; yellow) and state 1 in BtuB3G3-CNCbl (Fig. 6e; wheat). Moreover, state 2 density for CNCbl shows a different conformation for the vitamin (Supplementary Fig. 11b). These observations suggest that, after ligand capture by BtuG3, EL8 in the closed complex is poised to move inwards and destabilises the vitamin in its BtuG3 binding pocket, perhaps acting like a spring-loaded hinge. To test the importance of EL8, we generated a strain replacing the tip of EL8 (Ser524-Leu531) by two glycines in a *B. theta* background with loci 1 and 2 deleted (ΔEL8). We also replaced the native promoter, whose expression is repressed in the presence of $B_{12}$, with the strong constitutive P1E6 promoter[29] ensuring that both strains would have similar levels of expression even in the absence of vitamin import. Growth curves show that both strains can grow in the presence of methionine but the mutant cannot grow when methionine is replaced with $B_{12}$ (Fig. 6f). Moreover, both strains have similar amounts of BtuB3G3 in the OM (Fig. 6g). An analysis of EL8 from BtuB1, BtuB2 and BtuB3 shows that they are very similar in length and sequence, and that in both apo complexes (BtuB1G1 and BtuB2G2) the loop tip occupies the BtuG binding pocket (Supplementary Fig. 11f, g). Interestingly, Tyr538 in BtuB3G3-CNCbl (state1) hydrogen bonds with the nucleotide loop close to the phosphate from CNCbl (Fig. 6c). The aromatic character of this residue is

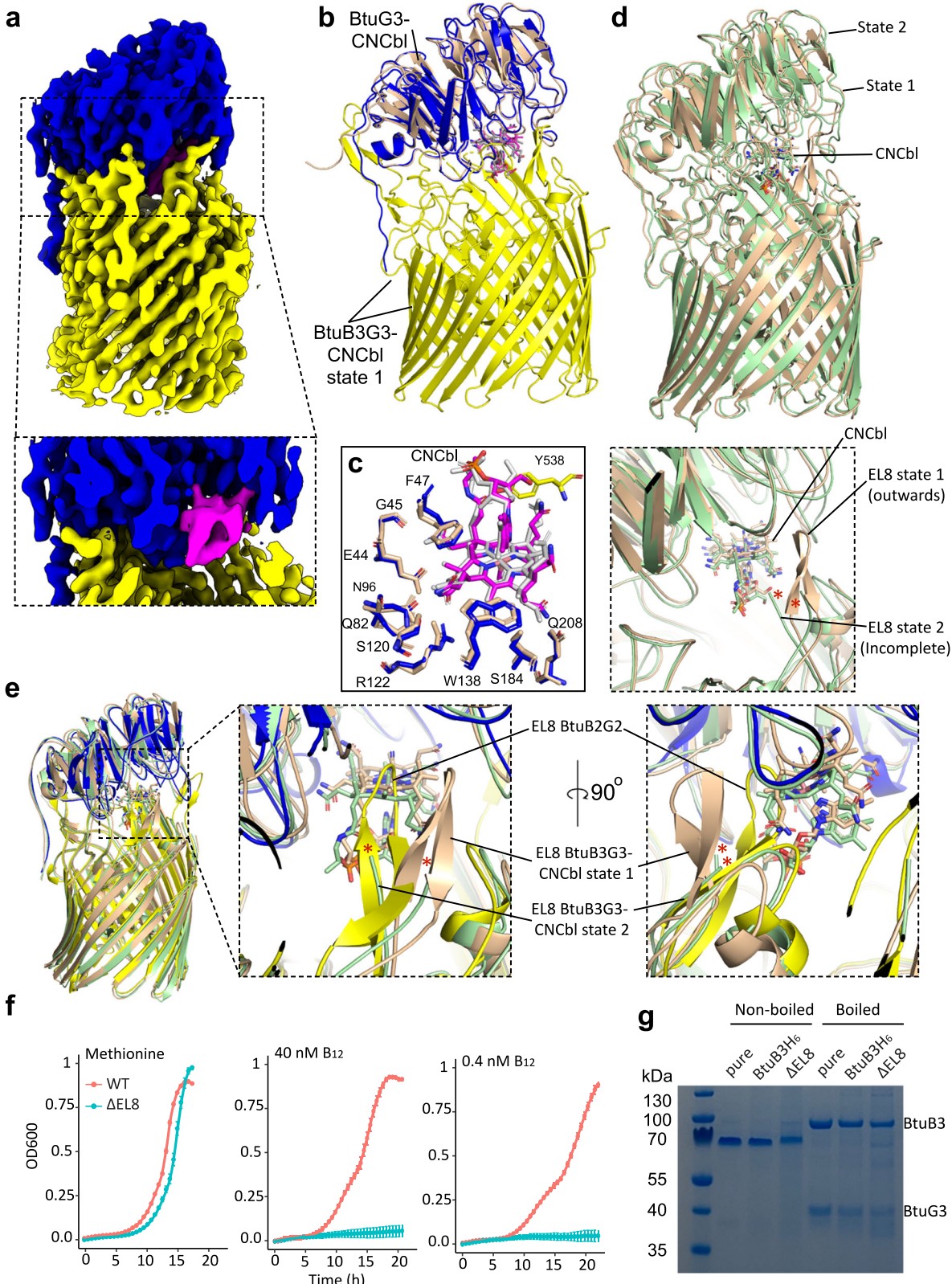

conserved in BtuB1 (Trp) and in BtuB2 (Phe), suggesting that the proximity of the electronegative aromatic ring could generate a repulsive force on the phosphate group of the bound $B_{12}$, enabling its dissociation from the BtuG binding site and translocation to BtuB. The lack of growth of the ΔEL8 mutant is most likely due to the extremely high affinity of CNCbl for BtuG, making dissociation and

translocation to BtuB a rare event in the absence of the destabilising EL8.

## CNCbl translocation from BtuG3 to BtuB3

The structural and functional data for the BtuB3G3-CNCbl complex, together with the comparison with the BtuG3-CNCbl and apo BtuB2G2

**Fig. 6 | Cryo-EM structure of BtuB3G3-CNcbl and functional analysis. a** Cryo-EM map for BtuB3G3-CNCbl state 1. BtuB3 is yellow, BtuG3 blue and CNCbl pink. **b** Cartoon superposition showing BtuB3G3-CNCbl state one (blue and yellow, vitamin in magenta) and BtuG3-CNCbl (wheat, vitamin in grey). **c** Close-up of the residues involved in vitamin binding by BtuG3 and BtuB3G3. Note that Y538 in EL8 (yellow) is the only BtuB3 residue that contacts the CNCbl molecule. **d** Top panel, superposition of BtuB3G3-CNCbl states 1 (wheat) and 2 (pale green). Bottom panel, close up view highlighting the differences in EL8. For clarity, loops are smoothened. The red asterisks denote the missing part of EL8 in state 2. **e** Same as **d**, but including the crystal structure of BtuB2G2 (in blue and yellow). Middle and right panel show close up views of EL8 and CNCbl. Note how CNCbl for BtuB3G3 state 2 (green) is slightly displaced towards BtuB3 relative to state 1. **f** Growth curves for WT (loci 1 and 2 deleted and a hexa-histidine tag in *btub3*) and EL8 mutant (WT plus the EL8 deletion; ΔEL8) strains in methionine, $B_{12}$-replete (40 nM CNCbl) and $B_{12}$-limiting conditions (0.4 nM CNCbl). Data are representative of three independent trials; error bars indicate ±SD from three technical replicates. Source data are provided. **g** Representative SDS-PAGE gel (n = 2 independent experiments) showing purified BtuB3G3 and IMAC elutions from WT and ΔEL8 strains. Note that the non-boiled samples show a stable complex, which after boiling dissociates in BtuB3 and BtuG3 (uncropped gels in Source Data).

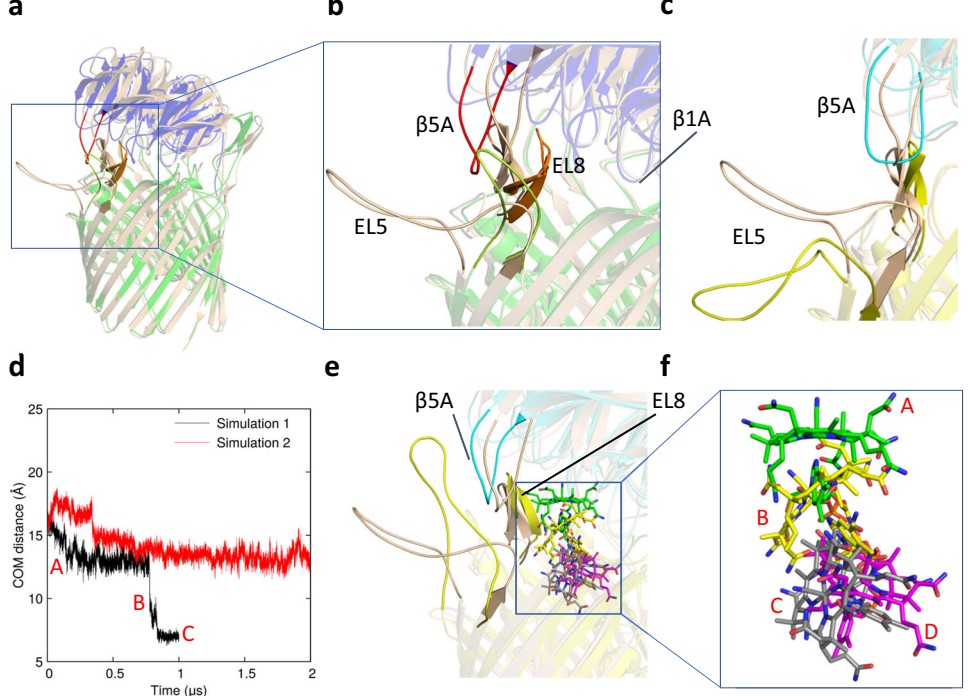

**Fig. 7 | BtuB3G3 structural dynamics and the CNCbl translocation process.**
**a** Comparison between the BtuB2G2 crystal structure (BtuB2 green and BtuG2 blue) and BtuB3G3 cryoEM structure (wheat), with vitamin $B_{12}$ not shown for clarity. **b** Close-up showing differences between the conformations of the EL8 loops of BtuB and of the β5A loops of BtuG (BtuB2 EL8, orange; BtuG2 β5A, red; BtuG2 EL5, green). **c** Loop rearrangements in BtuB3G3 before (wheat) and after (lime, BtuB3; cyan, BtuG3) an unbiased MD simulation in the absence of vitamin $B_{12}$. **d** Unbiased MD simulations based on the CNCbl-bound BtuB3G3 structure. Translocation of the CNCbl molecule was observed in simulation 1. Shown is the COM distance between CNCbl and the BtuB3 binding site. **e, f** Outline of the CNCbl translocation process during simulation 1. The initial structure of BtuB3G3 is in wheat, and the final structure is in lime (BtuB3) and cyan (BtuG3). Note the inward displacement of EL8 in BtuB3 and outward displacement of β5A in BtuG3 at the end of the simulation. CNCbl is coloured green in the BtuG3 binding site (A) and magenta in the BtuB3 binding site (D). An intermediate conformation from MD simulation 1 is in yellow (B), and the final conformation is in grey (C).

structures, suggest that the affinity of the CNCbl molecule for its BtuG3 binding site is lowered by the BtuB3 EL8 loop. To test this hypothesis, we performed unbiased MD simulations. First, we modelled the missing 24, 6 and 2 amino acid residues of BtuB3 loops EL5, EL4, and EL3 using MODELLER (10.4)[30]. Next, an unbiased MD simulation was performed after removal of vitamin $B_{12}$ to examine the dynamics of BtuB3G3, i.e. to ascertain whether it adopts a similar conformation to that of the apo BtuB2G2 crystal structure. Indeed, while the structural changes are fairly modest, upon removal of the $B_{12}$ molecule BtuB3G3 relaxes towards the BtuB2G2 structure, i.e. EL8 moves inwards towards the $B_{12}$ binding pocket (Supplementary Information Fig. 9h), while loop β5A of BtuG3 (involved in $B_{12}$ binding) moves outward, away from the $B_{12}$ binding pocket (Fig. 7a–c).

To generate the allosteric signal that exposes the TonB box of BtuB3 for interaction with periplasmic TonB, the substrate needs to be transferred from BtuG3 to the binding site in BtuB3. Therefore, we also explored whether CNCbl is transferred to BtuB3 via unbiased MD simulations (Supplementary movie 2). First, we docked CNCbl into

BtuB3 at a position very similar to that for *E. coli* BtuB (Supplementary Fig. 14 and Methods) and used this as a reference position[31]. As a measure for CNCbl movement, the centre of mass (COM) distance between CNCbl in the BtuB3 binding pocket and the initial position of CNCbl in BtuB3G3 was used. Remarkably, in one simulation (1), the CNCbl molecule dissociates from its binding site and moves -10 Å towards BtuB3 (Fig. 7d–f). In the other simulation (2), the ligand remains close to its BtuG3 binding site for 2 μs, illustrating the stochastic nature of ligand dissociation processes. At the end of simulation 1, the CNCbl molecule is close to the BtuB3 CNCbl binding site identified via docking, albeit with a different, tilted orientation (Fig. 7f) stabilised by electrostatic, CH-π, and π-π interactions with BtuG3 and BtuB3. The observation of CNCbl translocation in BtuB3G3 is in sharp contrast with simulations for BtuG2-CNCbl (Fig. 2) which all showed a strong attraction and stable binding of the vitamin to the isolated lipoprotein, in agreement with SPR data[19]. Once CNCbl has dissociated from its BtuG3 binding site, BtuB3 loop EL8 moves inwards (Supplementary Fig. 9i), but it does not reach the position inferred from the

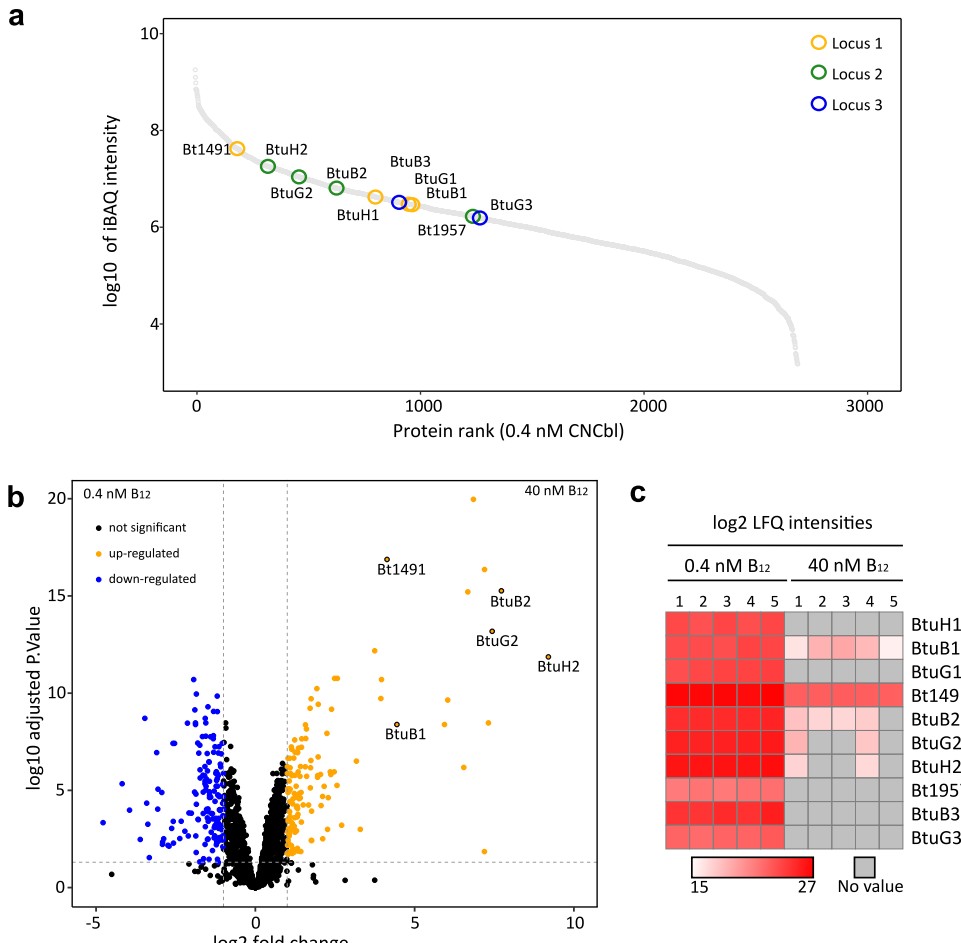

**Fig. 8 | Proteomic analysis of *B. theta* OM under B$_{12}$-restrictive (0.4 nM CNCbl) vs. B$_{12}$-replete conditions (40 nM CNCbl.). a** Relative abundance of OM proteins calculated using iBAQ intensities for cells grown in B$_{12}$-restrictive conditions. **b** Volcano plot showing the magnitude of change in the expression of OM proteins between B$_{12}$-restrictive vs. B$_{12}$-permissive conditions. Note that several proteins from the three loci are not detected under at least one condition. **c** Heatmap showing the LFQ intensities in both conditions, with each square representing an independent sample. Five independent cultures were grown in minimal media supplemented with 0.4 or 40 nM CNCbl for 18 h in anaerobic conditions. Grey indicates the protein has not been detected in this condition.

structures of apo BtuBG complexes, most likely due to the limited simulation time. Overall, however, the MD simulations confirm the importance of EL8 for the translocation of CNCbl from BtuG to BtuB.

**Quantitative proteomics of B$_{12}$ acquisition proteins**

An intriguing characteristic of many gut *Bacteroides* spp. is the presence of multiple B$_{12}$ acquisition loci. This raises the important question as to whether these systems are redundant or different, complementing each other in order to ensure optimal B$_{12}$ capture from the environment. An earlier study reported very different abilities of BtuB deletion strains to compete with wild type *B. theta* when B$_{12}$ is limiting. Strains encoding only BtuB1 or BtuB3 had large competitive defects, whereas a strain with only BtuB2 competed efficiently with wild type. In addition, cobalamins with different lower ligands selected for different single-BtuB strains in competition assays[18]. While these results could suggest different substrate specificities for the BtuB transporters, there is no information on OM levels of the various transporters (and indeed for any component of the three B$_{12}$ loci), and how those levels could be affected by different cobalamins. For example, the apparent efficiency of BtuB2 could easily be explained by a much higher level within the OM relative to BtuB1 or BtuB3. To start addressing this important question, we performed quantitative OM proteomics on log- and stationary phase wild type *B. theta* grown in B$_{12}$-limiting conditions (0.4 nM CNCbl) or B$_{12}$-replete conditions (40 nM CNCbl). As shown in Fig. 8a, components of all three loci are

present in the OM at low CNCbl concentration. Ranking the proteins by their relative label-free absolute protein quantification (iBAQ)[32] intensities, an approximation of absolute protein abundance, shows that under limiting B$_{12}$, BtuB2G2 is likely the most abundant BtuBG complex, followed by BtuB1G1. BtuB3G3 may be present at low numbers in the OM. These data suggest that the higher fitness of BtuB2[18] might be due to higher amounts in the OM. The recently described BtuH proteins, which are surface-exposed and also bind B$_{12}$ with high affinity[25] but do not stably associate with the BtuBG transporter core, appear to be more abundant than BtuB and BtuG. Interestingly, the uncharacterised Bt1491 protein of locus 1 is likely the most abundant protein by a considerably margin across all three loci under B$_{12}$-limiting and B$_{12}$-replete conditions, making it a priority to assess the potential role of this protein in B$_{12}$ acquisition. Its locus 2 paralog Bt1957, on the other hand, has a very low iBAQ ranking (Fig. 8a). Both Bt1491 and Bt1957 have signal sequences and a cysteine around position 20–30 followed closely by several acidic residues, consistent with lipoprotein export signal (LES) and surface exposure[33]. Finally, a comparison of the iBAQ ranking in B$_{12}$-limiting conditions vs. B$_{12}$-replete conditions suggests that all components of the three loci are much more abundant in the OM under limiting B$_{12}$ (Fig. 8b), as expected from transcriptomics data and the fact that the loci are under control of B$_{12}$-responsive riboswitches[18,34]. During B$_{12}$-replete conditions (40 nM CNCbl), expression of all Btu proteins is very much reduced, and most of the proteins are undetectable by mass spectrometry in OM samples

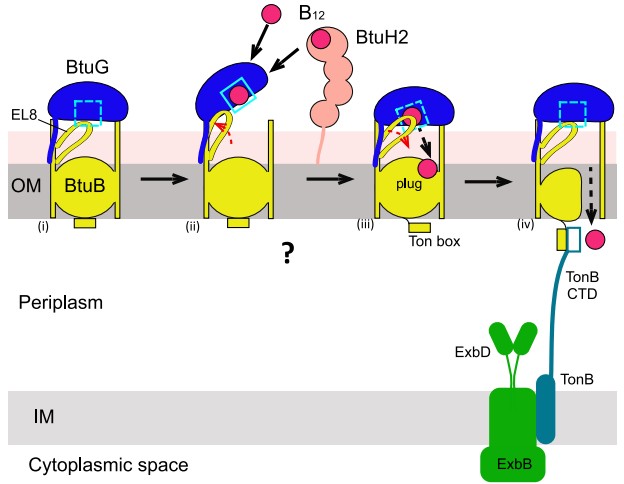

**Fig. 9 | Schematic model for lipoprotein-mediated B₁₂ acquisition by _B. theta_.**
Starting from the closed state in which EL8 occupies the $B_{12}$ binding site on BtuG, the BtuBG complex opens, which might happen spontaneously or perhaps is promoted by accessory proteins such as BtuH. (ii) After opening, EL8 moves away from the $B_{12}$ binding site, allowing acquisition of the vitamin by BtuG from the external environment or from BtuH. (iii) Upon lid closing, EL8 act as a spring-loaded hinge to destabilise the bound $B_{12}$, causing its release and transfer to BtuB. Binding of $B_{12}$ by BtuB generates allosteric changes in the plug that leads to TonB box exposure in the periplasmic space. During the final stage (iv), the C-terminal domain (CTD) of TonB binds to the TonB box and causes unfolding of the plug due to mechanical force generated by the TonB-ExbB-ExbD complex in the IM. The channel that is formed allows diffusion of the substrate into the periplasmic space[80].

(Fig. 8c). For future work, it will be interesting to assess the effect of different upper and lower ligands on OM protein levels, and to determine absolute copy numbers of the various proteins per cell.

## Discussion

How do the BtuBG transporters and additional surface-exposed $B_{12}$-binding lipoproteins such as BtuH, and possibly others, work together to take up corrinoids? The BtuBG transporters are structurally reminiscent of the widespread SusCD systems that mediate OM uptake of complex glycans[21,26]. Both types of complexes likely work in a similar fashion in that they consist of a TBDT with a mobile cap (SusD or BtuG), and use a "pedal-bin" mechanism for substrate loading and import[21,26]. However, in contrast to SusCD systems for which open and closed states are observed via cryo-EM, our data only show closed BtuBG complexes even in the absence of substrate. In addition, loading purified BtuBG with $B_{12}$ is challenging, again contrasting with SusCD systems that readily and stably bind substrate. Given the location of BtuG $B_{12}$ binding sites, it is clear that the BtuBG pedal bins will have to open for BtuG ligand capture. Apparently, the energy landscapes for lid opening are very different for BtuBG and SusCD systems, with deep minima for BtuBG closed states. One possibility is that the stably closed BtuBG complexes result from OM extraction and purification in detergent. It is conceivable, albeit speculative, that the LOS within the native OM environment, being highly negatively charged[28], could destabilise the closed state to favour opening. Alternatively, BtuG lid opening could be a relatively rare event, given that the $B_{12}$ requirements of the cell are likely to be modest, contrasting with SusCD systems which mediate uptake of carbon sources. Another possibility is that the additional $B_{12}$-binding surface lipoproteins, such as BtuH, are involved in loading the BtuBG complexes, e.g. via generation of the open transporter.

How are additional $B_{12}$-binding surface lipoproteins like BtuH organised relative to the core BtuBG transporter? Again, a comparison with SusCDs provides some insights. SusCs are atypical TBDTs and ~50% larger than regular TBDTs such as BtuB (~120 vs ~75 kDa). In a very

recent study, it was shown that one function of the increased SusC size is to provide binding interfaces for additional surface lipoproteins, in this case, glycoside hydrolases and surface glycan-binding proteins. Thus, multiple OM components of a polysaccharide utilisation locus (PUL) are organised within one large complex on the cell surface, termed utilisome[35]. BtuB proteins do not have the extra interfaces, and there is no room for stable association of other lipoproteins with the BtuBG core, which agrees with our IMAC pull downs in which only BtuB and BtuG co-purify, even in mild detergents. Thus, any interaction between BtuBG and other $B_{12}$-binding lipoproteins is likely to be transient. This then leads to a picture that contrast sharply with utilisomes, with $B_{12}$-binding lipoproteins in the OM that associate only transiently with their cognate BtuBG transporter. This also raises the question whether those lipoproteins could function in an "inter-locus" manner, e.g. could BtuH2 assist vitamin $B_{12}$ uptake by BtuB3G3?

Why would such differences in lipoprotein association exist? We speculate that this is due to the nature of the substrates and cellular requirements. Glycans are high molecular weight carbon sources that need to be processed into smaller fragments before being transported, while $B_{12}$ requires no processing. To ensure efficient capture of the processed glycans, it would make sense for the processing machinery (i.e. the additional lipoproteins) to be closely associated with the transporter, which would ensure relatively high turnover numbers. By contrast, the main purpose of $B_{12}$-binding surface lipoproteins would be to capture a relatively rare and valuable ligand without the need to be closely associated with the transporter to ensure high turnover. Indeed, the picomolar binding affinity of BtuH2[25] results in part from a very low $k_{off}$ ($\sim 8.6 \times 10^{-5}$ s⁻¹), corresponding to an average lifetime of the $B_{12}$-BtuH2 complex of several hours. Thus, on bacterial timescales, $B_{12}$ is bound virtually irreversibly and will likely be released only after interaction of BtuH2 with BtuB2G2. The low $k_{off}$ might also compensate for the low mobility of OMPs and LPS that emerges from studies in _E. coli_[36,37]. The fundamental difference in the nature of the substrate (polymers for SusCDs, a monomeric molecule for BtuBG) most likely explains why SusCD systems are dimers[21] and BtuBG monomers. Within the dimeric levan utilisome, the glycan binding sites of the surface glycan binding proteins are ~100 Å apart, and such dual "anchor points" might allow for more efficient processing of a polymer[35].

Free BtuG2 has an even higher affinity for CNCbl (~10⁻¹³ M) than BtuH2, and the average lifetime of the $B_{12}$-BtuG2 complex is about an hour[19], too long to be meaningful in vivo. So how does $B_{12}$ dissociate from its BtuG binding site to move to BtuB? Our experimental and computational data provide an elegant solution to this conundrum. After closure of the $B_{12}$-bound BtuG lid, a conserved extracellular BtuB loop (EL8) acts as a spring-loaded hinge to destabilise the vitamin in its binding site, resulting in transfer towards BtuB (Fig. 9). For SusCD systems, such conformational changes have not been observed and indeed may not be necessary since SusD lids bind their substrates with relatively low affinities (μM-sub mM). In conclusion, lipoprotein-capped TonB-dependent transporters seem to be exclusive for Bacteroidetes and most likely offer a competitive advantage in the gut. Previous data for SusCD complexes, added to our characterisation of the BtuBG systems of _B. theta_, suggest that the majority of _Bacteroides_ TBDTs function in a lipoprotein-assisted manner.

## Methods
### Culture conditions
_B. thetaiotaomicron_ was routinely grown on brain- heart infusion (BHI, OXOID) supplemented with hemin (1 μg/ml, Sigma-Aldrich). Agar (Fisher bioreagents), erythromycin (25 μg/ml, Duchefa Biochemies), gentamicin (200 μg/ml, Formedium) and 5-fluoro-2′-deoxyuridine (FUdR, 200 μg/ml, Sigma-Aldrich) were added when required. Minimal medium (MM) contained 7.5 mM $NH_4SO_4$ (Thermo Scientific), 9.5 mM $Na_2CO_3$ (Melford), 4 mM L-Cysteine (Melford), 100 mM KH2PO4

(pH 7.2)(Fisher Chemical), 1 µg/ml menadione (Sigma-Aldrich), 1 µg/ml hemin and a mixture of mineral salts (15 mM NaCl (Duchefa Biochemies), 0.18 mM CaCl$_2$•2H$_2$0 (Fisher Chemical), 0.1 mM MgCl$_2$•6H$_2$O (Fisher Chemical), 0.5 mM MnCl$_2$•4H$_2$O (Acros Organics) and 0.04 mM CoCl$_2$•6H$_2$O (Sigma-Aldrich)). Vitamin B$_{12}$ (Sigma-Aldrich) and L-methionine (Formedium) were added as required and described in the text. 0.5% of fructose (Thermo scientific) was added as carbon source. Cells were grown in a Whitley A35 anaerobic workstation with a mixture of gas of 80% N$_2$, 10% CO$_2$ and 10% H$_2$.

**Cloning, expression and protein purification.** The coding regions for the mature parts of *bt1954 (btuG2)* and *bt2095* (*btuG3*) were cloned into pET28 using NcoI and XhoI. The protein was expressed in BL21(DE3) in LB broth (Lennox L. Broth, Melford) 2.5 h at 37 °C with the addition of 1 mM IPTG (Melford) at OD ~ 0.6 to induce protein expression. Cells were collected by centrifugation, resuspended in TBS (10 mM Tris (Sigma-Aldrich), 300 mM NaCl; pH 8) lysed at a pressure of 20–23 kpsi using a cell disrupter (Constant Systems 0.75 kW), and purified by nickel affinity chromatography. Protein concentrations were measured using the BCA assay, and when needed, adenosylcobalamin, cyanocobalamin or cobinamide (all from Sigma-Aldrich) was added to a molar ratio 1:2 protein:corrinoid. The samples were incubated at 4 °C for 30 minutes and subjected to SEC using a HiLoad 26/600 Superdex 200 column (GE healthcare).

**Overexpression and purification of BtuBG complexes.** To a strain with loci 1 and 3 deleted[18], a 6 x his-tag was added to the C terminus of genomic *bt1953* (*btuB2*) using pExchange-*tdk*. The expression of locus 2 was driven by P1E6[29] and not by the B$_{12}$-dependent wild-type riboswitch. The mutated strain was grown for 20 h in minimal media with 0.5% fructose under anaerobic conditions and collected by centrifugation at 11,000 × *g* for 30 min. Pellets were resuspended in TBS and lysed at a pressure of 23 kpsi using a cell disrupter (Constant Systems 0.75 kW). The membrane fraction was harvested by centrifugation (45 minutes at 234000 g. using a 45 Ti Beckman rotor). The inner membrane from 4 litres was solubilized in 0.5% sodium *N*-lauroyl sarcosine in 20 mM HEPES pH 7.5 and the sample was centrifugated for 30 minutes at 234000 g in a 45 Ti Beckman rotor. The pellet was solubilized in TBS plus 1.5% n-dodecyl-*N,N*-dimethylamine-*N*-oxide (LDAO; anatrace) 1 h at 4 °C and centrifuged again. The supernatant was loaded onto a nickel column (TBS 0.2% LDAO 25 mM imidazole) and the eluted sample (250 mM imidazole, Fisher Chemical) was further purified by SEC (HiLoad 26/600 Superdex 200 GE healthcare) using 10 mM HEPES pH 7.5, 100 mM NaCl and 0.05% LDAO (Anagrade, Anatrace). The eluted samples from 10 purifications (~2.5 mg total) were pooled together and buffer-exchanged into 10 mM HEPES pH 7.5, 100 mM NaCl and 0.4% tetraethylene glycol monooctyl ether (C$_8$E$_4$, Anatrace). Protein for crystallisation was concentrated to ~8 mg/ml, aliquoted and flash-frozen in liquid nitrogen. All protein samples were stored at −80 °C.

For BtuB1G1, we introduced a C-terminal hexa-histidine tag into *BT1489* (*btuB1*) in a *B. theta* mutant strain lacking loci 2 and 3[18]. We then replaced the wild-type promoter (B$_{12}$-dependent riboswitch) with the constitutive promoter P1E6 to obtain high levels of expression. Growth conditions were similar to BtuB2 but the minimal media was supplemented with 0.25 mM L-methionine (as the strain lacks both inner membrane B$_{12}$ ABC transporters). Five 4l batches of cells were purified in the same way as for BtuB2 with a minor modification. The final SEC column was run using 10 mM HEPES pH 7.5, 100 mM NaCl and 0.05% dodecyl-maltoside (DDM, Anatrace). The yield obtained was ~50 micrograms/l.

For BtuB3G3, we introduced a C-terminal hexa-histidine tag into *BT2094* (*btuB3*) in a *B. theta* mutant strain lacking loci 1 and 2[18]. We then replaced the wild-type promoter (B$_{12}$-dependent riboswitch) with the constitutive promoter P1E6[29] to obtain high levels of expression.

Growth conditions were similar to BtuB2. The pellet (4 L) was resuspended in TBS and lysed at a pressure of 23 kpsi using a cell disrupter (Constant Systems 0.75 kW). The membrane fraction was harvested by centrifugation (45 minutes at 234000 g. using a 45 Ti Beckman rotor). The inner membrane was solubilized in 0.5% sodium *N*-lauroyl sarcosine (Amresco) in 20 mM HEPES (Fisher Chemical) pH 7.5 and the sample was centrifugated for 30 minutes at 234000 g in a 45 Ti Beckman rotor. The pellet was solubilized in TBS plus 1.5% n-dodecyl-*N,N*-dimethylamine-*N*-oxide (LDAO anatrace) 1 h at 4 °C and centrifuged again. The supernatant was loaded onto a nickel column (TBS 0.05% LMNG (Lauryl Maltose Neopentyl Glycol, anatrace) 25 mM imidazole) and the eluted sample (250 mM imidazole) further purified by size exclusion chromatography (HiLoad 16/600 Superdex 200; GE healthcare) using 10 mM HEPES pH 7.5, 100 mM NaCl and 0.02% LMNG (anagrade). About 600 µg of purified BtuB3G3 were incubated with 0.5 mM CNCbl for 48 hours at 20 °C, and then excess CNCbl was removed by size exclusion chromatography (HiLoad 26/600 Superdex 200 GE healthcare) using 10 mM HEPES pH 7.5, 100 mM NaCl (NB no LMNG added) and the final samples were pooled together and concentrated to ~4 mg/ml for cryo-EM.

For functional analyses, 1 litre each of WT and EL8 mutant was grown in permissive B$_{12}$ conditions to an optical density at 600 of 0.6. Cells were harvested by centrifugation and to manipulate as little as possible the pellet, sonication was the method preferred for cell lysis. The pellet was sonicated for 3 minutes and processed as described above. For both cultures all conditions during growth and purification were the same. The same elution volume from the nickel column was loaded into an SDS-Page to be able to compare the level of expression.

## Growth curves

Overnight cultures of the desired strains were grown overnight under anaerobic conditions in BHI supplemented with hemin (1 µg/ml), and subcultured for 4 h next morning in fresh media. Cells were washed two times with minimal media without B$_{12}$, The OD600 was adjusted to 0.03 and 100 µl of cells were dispensed into wells of a 96-well plate. Negative control wells (with no B$_{12}$ nor L-methionine), positive control wells (with 200 mM L-methionine) and cultures with permissive and restrictive conditions for B$_{12}$ were set up in triplicates and grown for 24 h in the anaerobic cabinet. OD measurements were done using a Biotek Epoch microplate reader. Data was collected using Biotek Gen5 (2.09 Agilent) software and analysed and plotted with the R package (Version 4.2.3) Ggplot2[38] (Version 3.4.2), standard deviation bars are represented in plots.

## Crystallisation and X-ray crystal structure determination

Sitting drop vapour diffusion crystallisation trials were set up with a Mosquito Crystallization robot (TTP Labtech) using the commercial screens MemGold and MemGold2 (MG and MG2; Molecular Dimensions) for BtuB2-BtuG2 (in the presence of 1.5-fold molar excess of CNCbl) and JCGS+, Structure (Molecular Dimensions) and Index (Hampton research) for BtuG2. For BtuG2-CNCbl and BtuG2-AdoCbl we obtained crystals in condition E4 from JCSG+ (0.2 M Lithium sulphate, 0.1 M Tris pH 8.5, 1.26 M ammonium sulphate). These were cryoprotected with saturated ammonium sulphate. For BtuG2-Cbi, crystals were obtained in C9 from Index (1.1 M Na Malonate pH 7, 0.1 M HEPES pH 7 and 0.5% Jeffamine ED2001 pH 7) and cryoprotected with 20% PEG400 (Molecular dimensions). In all cases, ligand was added at ~1.5-fold molar excess relative to protein and incubated at room temperature for ~1 hr prior to setting up crystallisation trials. For the BtuB2G2 complex, the initial hit C2 from MG2 (0.08 M magnesium acetate tetrahydrate, 0.1 M sodium citrate pH 6.0, 14% PEG 5000 MME) was further optimised with variations of 2% of the original PEG concentration. The crystals were cryoprotected using 23% PEG400 and flash frozen in liquid nitrogen. Diffraction data were collected at 100 K at Diamond Light Source (supplementary Table 1). Dataset for the

Co-SAD experiment was integrated using XDS[39] (Feb 5, 2021), Pointless[40] (1.12.14) was used for space group determination and then scaled and merged with Aimless[40,41] (0.7.9). To find the heavy atom substructure, Phenix AutoSol[42,43] was used. Clear electron density was visible for BtuG2 and for a molecule of B[12]. An initial model was built using AutoBuild[44] in Phenix (1.20.1-4487) and manual building in Coot[45] (0.9.8.8). The high-resolution data set for BtuG2-CNCbl was integrated with Dials[46] (3.11.1), while BtuG2-AdoCbl and BtuG2-Cbi were integrated with XDS[39]. These three datasets were scaled and merged using Aimless[40,41]. The initial model produced from the Co-SAD experiment was used to solve the phase problem by molecular replacement with Phaser[47]. All the models were improved by rounds of manual building using Coot[45] and refinement using Refmac 5.8.0267[48] in CCP4 cloud[49] for BtuG2-AdoCbl and BtuG2-CNCbl and Phenix refine[50] for the Co-SAD experiment and BtuG2-Cbi. Models for CNCbl (CNC in REFMAC monomer library) AdoCbl (B1Z) and Cbi (CBY) were placed in coot. For the high-resolution model, anisotropic refinement with automated local NCS (non-crystallographic symmetry) restrains were used while isotropic refinement with TLS and automated local NCS restrains were selected for BtuG2-AdoB[12]. Crystals for BtuG3-CNCbl were obtained similarly to BtuG2-CNCbl (Index H8; 0.1 M Magnesium Formate, 15% PEG 3350). A data set was integrated with XDS[39] and scaled, merged and cut to a resolution of 2.6 Å using Aimless[40,41]. The phase problem was solved via molecular replacement with Phaser[47] using BtuG2-CNCbl as search model. The model was improved by rounds of manual building using coot[45] and refinement using Refmac 5.8[48] in CCP4 cloud[49]. Diffraction data for the BtuB2G2 complex was integrated with XDS[39], the space group was determined with Pointless, then scaled and merged with Aimless[40,41]. The phase problem was solved with a two-component search in Phaser[47] using the models for BtuG2 and 2GUF (BtuB from *E. coli* after modification by Sculptor within Phenix). The model was built with iterative cycles of AutoBuild[44] and manual building in Coot[45]. The model was refined in Phenix using secondary structure and BtuG2 model restrains, NCS and TLS. MolProbity[51] (4.2) was used to validate protein geometry and PyMol was used for the visualisation of the protein structures. To calculate the evolutionary conservation of the amino acid positions, the phylogenetic relations of 150 homologous sequenced were analysed and colour coded according to their conservation value using the ConSurf server[23]. The multiple sequence alignment was built using MAFFT, collecting the homologues from UNIREF90. The homologue search algorithm used was HMMER (HMMER-value: 0.0001) with 1 iteration and a maximal and minimal percentage of identity of 95% and 35% respectively. The chains used for the analysis were C and D from the BtuB2G2 model, since those had the best electron density.

## Cryo-EM structure determinations

Purified BtuB1G1 complex (~2.5 mg/ml) without CNCbl was applied to glow-discharged Quantifoil 0.6/1 300 mesh holey carbon grids, blotted and plunge-frozen in liquid ethane using a Vitrobot Mark IV device (Thermo Fisher Scientific). Grids were imaged on a FEI Titan Krios microscope (Thermo Fisher Scientific) operating at 300 kV (Supplementary Table 2). Movies were recorded using EPU (Thermo Fisher Scientific, v3.2) on a Falcon 4 direct electron detector in electron event representation (EER) mode at 130,000 magnification, corresponding to a pixel size of 0.91 Å. The microscope was equipped with a Selectris X energy filter with slit width set to 5 eV. 1924 movies were recorded in total. Data were processed in cryoSPARC v3.3.2[52] (Supplementary Table 2). Following patch motion correction and patch CTF correction, ~2000 particles were manually picked and used to make templates for template-based picking. 1,110,283 particles were extracted in 300 × 300 pixel boxes and subjected to two rounds of 2D classification. Good classes were selected and used to make an ab initio model using a stochastic gradient descent approach with 2 classes. The class showing secondary structure features was subjected to non-uniform

refinement[53]. Several rounds of heterogeneous refinement, non-uniform refinement and 2D classification were used to discard the remaining bad particles, resulting in the final particle stack with 50,547 particles. The particles were re-extracted in 384 × 384 pixel boxes and subjected to a final round of non-uniform refinement with defocus and global CTF refinement enabled, resulting in a 3.22 Å reconstruction. An ab initio model was built into the cryo-EM map using Buccaneer (part of the CCPEM package[54]) followed by iterative manual building in Coot[45] and real space refinement in phenix[55]. The initial atomic model for BtuB1 and the G domain of BtuG1 was used to sharpen the density using LocScale[56] to improve interpretability and aid with model building. To build the H domain of BtuG1, the final reconstruction from non-uniform refinement was sharpened with a B-factor of −50 Å². The BtuB1G1 model built into the LocScale sharpened map was refined into the B-factor-sharpened map together with the H domain of BtuG1. There is strong density extending from the side chain of T164 located on the β-propeller of BtuG1, which likely corresponds to the presence of a glycan moiety. The precise composition of the O-glycan is unknown, therefore a single β-D-galactopyranosyl modification was modelled. The refinement statistics of the final model can be found in Supplementary Table 2.

Purified BtuB1G1 complex with 2 equivalents CNCbl was applied to glow discharged Quantifoil 0.6/1 300 mesh holey carbon grids, blotted and plunge frozen using a Vitrobot Mark IV device. The grids were imaged on a FEI Titan Krios microscope operating at 300 kV. Movies were recorded on a Falcon 3 direct electron detector in counting mode at 75,000 magnification, corresponding to a pixel size of 1.065 Å. 626 movies were recorded in total. Data were initially processed as above, but following 2D classification no 3D reconstruction with secondary structure features could be obtained from ab initio reconstructions or non-uniform refinement.

Purified BtuB3G3 complex in LMNG and loaded with CNCbl (4 mg/ml) was applied to glow-discharged Quantifoil 1.2/1.3 200 mesh holey carbon grids, blotted and plunge-frozen in liquid ethane using a Vitrobot Mark IV device (Thermo Fisher Scientific). Grids were imaged on a FEI Titan Krios microscope (Thermo Fisher Scientific) operating at 300 kV (Supplementary Table 2). Movies were recorded using EPU (Thermo Fisher Scientific) on a Falcon 4i direct electron detector in electron event representation (EER) mode at 165,000 magnification, corresponding to a pixel size of 0.74 Å. The microscope was equipped with a Selectris X energy filter with slit width set to 10 eV. 9826 movies were recorded in total. Data were processed in cryoSPARC[52] v4.1.2 (Supplementary Table 2 and Supplementary Fig. 12). Following patch motion correction and patch CTF correction, ~1600 particles were manually picked and used to make templates for template-based picking. 2,110,683 particles were extracted in 300 × 300 pixel boxes (1.48 Å/pix) and subjected to two rounds of 2D classification. 200,000 particles were used in ab initio reconstruction with 5 classes. The full 497,190 particle stack was subjected to heterogeneous refinement against the 5 ab initio models. The class exhibiting protein-like features was subjected to non-uniform refinement, resulting in a reconstruction with an estimated resolution of 3.8 Å but poor secondary structure features. Further 2D classification of this particle stack revealed subsets of monomeric and dimeric particle populations (Supplementary Fig. 12). It is unlikely that the dimerization is of biological relevance as the two copies of BtuB3G3 are rotated ~150° relative to each other, and the TonB box of one BtuB3 copy would face the extracellular space. The dimerization is likely the result of the chosen detergent. Monomer and dimer maps reconstructed by non-uniform refinement were used to generate templates for particle picking. The whole dataset was separately picked using the monomer and dimer templates. After removing duplicates and 2D classification, 804,932 particles remained. This particle stack was subjected to heterogeneous refinement using 6 classes (monomer, dimer and 4 decoys).

Non-uniform refinement of the dimer class (219,984 particles) resulted in a 3 Å reconstruction. Further 3D classification focused on BtuG3 and the extracellular portion of BtuB3 (see Supplementary Fig. 12 for mask) distinguished two sub-populations with differences in the EL8 and CNCbl region, state 1 and state 2. After particle re-extraction (512 pix box, 0.74 Å/pix sampling rate), non-uniform refinement with enabled per-particle defocus and CTF (beam tilt and trefoil) refinement, and local refinement, reconstructions of state 1 (66,318 particles) and state (91,067 particles) were obtained at global resolutions of 2.97 Å and 2.75 Å, respectively (Supplementary Fig. 12). Both state 1 and state 2 reconstructions showed slight preferred particle orientation but this did not result in significant distortion of the maps (Supplementary Fig. 13 showing particle distribution plots and 3DFSC), as confirmed by 3DFSC analysis[57]. AlphaFold2[58] models of BtuB3 and BtuG3 were docked into one BtuB3G3 copy in the sharpened dimeric cryo-EM maps in phenix[59] and manually adjusted in Coot[45]. CNCbl was manually placed in Coot. Models were subjected to cycles of manual building in Coot[45] and real space refinement in phenix[55]. Model validation was performed using MolProbity[51]. The poor density of CNCbl and EL8 could not be improved by focused 3D classification of the monomer maps, and model building into the monomer map was not performed (Supplementary Fig. 12).

## MD simulation methods

All BtuB2G2 and BtuB3G3 systems for the MD simulations in the presence and absence of cyanocobalamin were built using the CHARMM-GUI Membrane Builder[60]. Using the PROPKA webserver, the protonation states of the titratable amino acids were verified to be in their standard protonation states. The complex was placed into a 1-palmitoyl 2-oleoyl phosphatidyl-ethanolamine (POPE) bilayer and solvated with TIP3P water molecules on both sides of the membrane maintaining a water thickness of about 25 Å. The main reason for using a POPE bilayer is that the OM composition of *B. theta* is far from clear. Even though the *B. theta* OM is likely to be asymmetric and contains highly negatively charged lipooligosaccharide (LOS) molecules in the outer leaflet[28], there is no data on the extent of charge neutralisation via divalent metal ions as in Proteobacteria. Moreover, *B. theta*, like other *Bacteroides*, is known to contain abundant sphingolipids[61–63] and there is no reason to assume that these won't be present in the OM. Thus, while using a POPE bilayer for the *B. theta* OM is undoubtedly inaccurate, using an *E. coli*-derived asymmetric OM model with rough LPS is likely to be non-physiological as well. In addition, our main goal was to characterise the B$_{12}$ acquisition process and the translocation to BtuB. The inclusion of LPS/LOS molecules in the membrane would have led to additional convergence issues due to the known, very slow diffusion of these molecules in the membrane. Combined with the fact that it is not known which BtuB loops interact with LOS/LPS, such studies fall beyond the scope of the current manuscript. Since demineralized water is a typical artefact of MD simulations and as salt can greatly influence protein stability and conformations, we studied the proteins under physiological conditions of 0.15 M KCl by adding K$^+$ or Cl$^-$ ions to achieve the desired concentration and to neutralise the system.

The total system composed of the protein, membrane, solvent, ions, and ligand was placed into a rectangular box of $80 \times 80 \times 150$ Å$^3$. For simulations in the absence of BtuB2, the surface-exposed lipoprotein BtuG2 was simulated in an aqueous environment after neutralising the 18*e* negative charge with K$^+$ ions and in the presence of additional 0.15 M KCl. The system composed of BtuG2, water, ions and ligand was placed into a rectangular box of size $72 \times 88 \times 77$ Å$^3$. All simulations were conducted using the GROMACS molecular dynamics software, version 5.1.4, and employing the CHARMM36-m forcefield[64,65].

Energy minimisation for the model systems was performed using the steepest descent method with 50.000 steps, while the systems were well minimised before reaching that limit. This step was followed by a two-step constant volume (NVT) equilibration of 5 ns each with varying restrains on the simulated system at 300 K. The NVT simulations were carried out using a Berendsen thermostat with a 1-picosecond temperature coupling constant and the calculated average system temperature was 300 K. Furthermore, the systems were relaxed in a four-step constant pressure (NPT) equilibration of 50 ns by removing the restraints on the protein, the membrane, and the ligands in a stepwise manner. The NPT simulations were carried out using the semi-isotropic coupling method to a Berendsen barostat at one bar with a coupling constant of 2 ps. For non-bonded interactions, the Verlet cut-off scheme for the Coulomb and Lennard-Jones interactions was employed with a cut-off of 10 Å. Moreover, the Particle Mesh Ewald scheme was used for evaluating the long-range electrostatic interactions, and all bonds in the proteins were constrained using the Linear Constraint Solver (LINCS) algorithm[66,67]. After the six-step equilibration, the final simulations were continued according to the protocols defined for unbiased simulations. The NPT production runs were performed using a Parrinello-Rahman[68] barostat along with a semi-isotropic pressure coupling and the Nose-Hover thermostat[69] (unbiased and metadynamics MD simulations).

To understand the system stability and structural changes during the permeation of cyanocobalamin through the BtuBG protein, we analysed the root mean square deviations from the initial structure (RMSD) and the radius of gyration (R$_g$) of the proteins (data not shown). Further angle analyses were performed to examine the extent of lid opening of the BtuB2G2 protein complex via a hinge loop. The short-range electrostatic interaction between the CNCbl and BtuG2 was calculated using the '*gmx energy*' tool.

## Free energy calculations

To determine the free energies of biomolecular processes, the metadynamics (MtD) technique was employed that progressively builds up a history-dependent biasing potential along predefined collective variables (CVs)[70]. The sum of these Gaussian potentials is given by[71]

$$V(s,t) = \sum_{t_i} H \exp\left(-\frac{|s - s(t_i)|^2}{2w^2}\right) \tag{1}$$

where $H$ denotes the height of the Gaussian hills, $w$ their width, and s the CV. In a MtD simulation, the bias or "hills" are dynamically placed on top of the underlying potential energy landscape and discourage the system from re-visiting the same points in configurational space. This extra bias enforces, e.g., an unbinding process, within a reasonable computational time. In the well-tempered version of metadynamics, which is used in the study, one rescales the Gaussian height with the bias accumulated over time at a fictitious higher temperature, T + ΔT[72]. Despite its inherent nonequilibrium characteristics, one can extract information close to the true equilibrium state of the system by suitably tuning the parameter ΔT. The free energy profile usually termed potential of mean force (PMF) can be estimated as

$$F(s,t) = -\frac{T+\Delta T}{\Delta T}V(s,t) + k_b T \log \int \exp\left(\frac{T+\Delta T}{k_b T \Delta T}\right)V(s,t)\,ds \tag{2}$$

The centre of mass between the CNCbl and the BtuG2 binding cavities name z was considered as the major CV for the binding free energy calculation between the CNCbl and BtuG2. The corresponding dissociation constant was calculated based on the free energy difference ΔG was estimated using $k = e^{-\Delta G/RT}$ at 300 K. To construct the 2D free energy surface, a second CV was defined as the projection $z_{ij}$ of the distance $r_{ij}$ between two porphyrin carbon onto the z axis (Supplementary Fig. 3c). The orientation of the substrate molecule can be determined as $\varphi = \cos^{-1}(r_{ij}/z_{ij})$, i.e., the value of $z_{ij}$ determines the orientation of the molecule. To examine the free energy associated with the BtuG2 lid opening process, multiple-walker WTMTD

simulations were carried out. The 1D free energy profile was estimated as a function of the CV angle, which can depict the aperture between BtuG2 and BtuB2, between amino acids group Leu467-Gly469 of BtuB2, Asp36-Gly38 from the linker of BtuG2, and Thr235-Gly237 from the junction of a flexible and stable loop of BtuG2.

## Sample preparation and mass spectrometry for proteomics

Five independent cultures were grown in minimal media with 0.4 nM or 40 nM CNCbl and grown for 18 h. Cells were collected by centrifugation at $11,000 \times g$ for 30 min. The pellets were resuspended in TBS and sonicated. The membrane fraction was harvested by centrifugation (45 min at $234,000 \times g$. using a 45 Ti Beckman rotor). The fractions were washed two times with water (resuspended in water followed by 45 min at $256,000 \times g$ using a 70 Ti Beckman rotor). Finally, they were stored at −80 °C until needed.

Membrane fractions were subjected to denaturation and tryptic digest using a suspension trapping-S-trap (Protifi) protocol. Briefly, this included resuspension in 5% SDS, 50 mM Tris pH 7.4, denaturation with 5 mM tris(2-carboxyethyl)phosphine (TCEP) at 60 °C for 15 min, alkylation with 10 mM $N$-ethylmaleimide (NEM) at room temperature for 15 min, and acidification to a final concentration of 2.7% phosphoric acid. Samples were then diluted eightfold with 90% MeOH 10% TEAB (pH 7.2) and added to the S-trap micro columns. The manufacturer-provided protocol was then followed, with a total of five washes in 90% MeOH 10% TEAB (pH 7.2), and trypsin added at a ratio of 1:10 enzyme:protein (2 µg: 20 µg) and digestion performed for 2 h at 47 °C. Peptides were dried and stored at −80 °C, and immediately before mass spectrometry were resuspended in 0.1% formic acid. LC-MS/MS was performed using an Ultimate 3000 RSLCnano System (ThermoFisher Scientific) in line with an Orbitrap Fusion Lumos Tribrid mass spectrometer (ThermoFisher Scientific). Peptides (1 µg) were injected onto a PepMap100 C18 LC trap column (300 µm ID x 5 mm, 5 µm, 100 Å), and separated with an EASY-Spray nanoLC C18 column (75 µm ID × 50 cm, 2 µm, 100 Å) at a flow rate of 250 nl/min, column temperature 45 °C. Solvent A was 0.1% (v/v) formic acid in HPLC water, and solvent B was 0.1% (v/v) formic acid and 80% (v/v) acetonitrile in HPLC water. LC-MS/MS runs were preceded by a 2-min equilibration with solvent A and solvent B at 98% and 2%, which was maintained for 5 min following sample injection, then the gradient increased solvent B to 35% over the next 120 min. Solvent B was increased to 90% in 30 s for 4 min, then decreased to 2% in 30 s to allow further equilibration for 10 min. Data were acquired by the Orbitrap Fusion Lumos in positive ion mode, with data-dependent acquisition (software Xcalibur 4.4 (ThermoFisher Scientific)). MS1 was performed at 120,000 resolution, in the scan range 400–1600 $m/z$, charge states 2-7, AGC target of 200,000, with a maximum injection of 50 ms with repeat count 1. Peptides were fragmented using HCD (30% collision energy). The ion trap was selected as the detector type, set to rapid scan rate, mass range and scan range mode set to normal and auto, AGC target set to standard, maximum injection time set to dynamic. The entire duty cycle lasted 3 seconds, during which time "TopSpeed" analysis was performed.

## Proteome data analysis

Raw files from mass spectrometry were matched against the *B. thetaiotaomicron* reference proteome (UP000001414, downloaded from UniProt on 01/12/2021), and proteins quantified using MaxQuant V 2.0.3.0[73]. NEM alkylation of cysteine was set as a fixed modification, oxidation of methionine and acetylation of protein N-termini were set as variable modifications. Digestion was trypsin/p specific, LFQ and matching between runs was selected. All other parameters were left as defaults; mass tolerance for precursor and fragment ions was 20 ppm, minimum peptide length was 7 residues, and 1% FDR for peptide- and protein-identification was used. The output from MaxQuant was processed in Perseus (2.0.3.1)[74], where contaminants and decoys were removed and iBAQ was used to rank protein expression within each condition. Processing was also performed using the Limma package[75] in the R programming environment, where LFQ intensity values were used to quantify relative protein abundance between different conditions. Proteins were filtered to remove contaminants and decoys, and those identified by <2 unique peptides. Changes in protein abundance between conditions were considered significant where there was a difference of at least twofold, and when Student's T-test $p$-value =<0.05 after Benjamini-Hochberg correction for multiple comparisons. We used the R package Ggplot2[38] to generate proteomic analysis graphs. Morpheus (https://software.broadinstitute.org/morpheus) was used to generate the heatmap.

## Reporting summary

Further information on research design is available in the Nature Portfolio Reporting Summary linked to this article.

## Data availability

The data supporting the findings of this study are available upon the corresponding authors upon reasonable request. The mass spectrometry proteomics data have been deposited to the ProteomeXchange Consortium[76] via the PRIDE partner repository[77] with the data set identifier: PXD038230. For X-ray structures coordinates and structure factors have been deposited in the Protein Data Bank with accession codes 8BMX for BtuG2-CNCbl, 8BMY for BtuG2-AdoCbl, 8BZM for BtuG2-Cbi, 8BM0 for BtuB2G2 and 8OKV for BtuG3-CNCbl. EM structure coordinates have been deposited in the Protein Data Bank and EM maps in the Electron Microscopy Data Bank with accession codes 8BLW and EMD-16114 for BtuB1G1, 8P98 and EMD-17575 for BtuB3G3-CNCbl state1 and 8P97 and EMD-17574 for BtuB3G3-CNCbl state2. The pdb model used for molecular replacement was previously deposited as 3DSM. Initial and final conformations for the molecular dynamics analysis have been deposited in the Zenodo repository (https://doi.org/10.5281/zenodo.8164805). Source data are provided with this paper.

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

## Acknowledgements

The research of B.v.d.B. is supported by a Wellcome Trust Investigator award (214222/Z/18/Z), providing salary support for J.A.R. and A.S. We would like to acknowledge the Diamond Light Source for crystal-lography beam line access (proposal mx-24948) and i04 beamline support, Dr Dan Maskell and the Astbury Biostructure Laboratory (Leeds) for collecting cryo-EM data, the Newcastle Structural Biology Laboratory for data processing infrastructure and Professor Andrew Goodman (Yale University) for providing *Bacteroides thetaiotaomicron* strains. J.A.R. would like to acknowledge Jonathan Drury for his assistance with plots. K.J. acknowledges the Alexander von Humboldt (AvH) foundation for an AvH postdoctoral research fellowship. K.J. and U.K. are grateful to the North German Supercomputing Alliance (Norddeutscher Verbund für Hoch- und Höchstleistungsrechnen – HLRN) for providing access to their high-performance computational facilities through project hbp00058. K.J. thanks Dr. Jigneshkumar Dahyabhai Prajapati and Dr. Vinaya Kumar Golla for helpful scientific discussions. A.M.F. and M.T. are funded by a Wellcome Investigator Award to M.T. (215542/Z/19/Z).

## Author contributions

J.A-R. and B.v.d.B. expressed and purified proteins and determined X-ray crystal structures. A.S. determined the cryo-EM structures. J.A.-R. per-formed the functional analysis. K.J. and U.K. carried out molecular dynamic simulations. A.B. maintained the Newcastle Structural Biology Laboratory. B.v.d.B and U.K. supervised the structural and computa-tional studies, respectively. J.A.-R., K.J., U.K. and B.v.d.B. contributed to writing the manuscript. A.M.F. performed and analysed the proteomic experiments. M.T. supervised the proteomic studies.

## Competing interests

The authors declare no competing interests.

## Additional information

**Peer review information** *Nature Communications* thanks Konstantinos Beis, Dongchun Ni and the other, anonymous, reviewers for their con-tribution to the peer review of this work. A peer review file is available.

