## [Peer Review File · Nature Communications]

REVIEWER COMMENTS

Reviewer #1 (Remarks to the Author):

The work of Abellon-Ruiz et al. focuses on the initial phase of the TonB-dependent transport of vitamin B12 in the Bacteroides bacteria of the gut. The authors use a variety of methods such as X-ray crystallography, cryo-EM, proteomics analyses and molecular dynamics (MD) simulations to determine the binding of vitamin B12 to the BtuG receptor and its transfer to BtuB.

The gut B. theta species have several vitamin B12 receptors in the outer-membrane, contrary to E. coli. In addition, they encode for lipoproteins that capture and bind the scarce vitamin B12 prior to delivering it to the outer-membrane receptor BtuB. The authors focused on the structure, interactions and vitamin B12 transfer between the BtuG2 lipoprotein scavenging B12 and the BtuB2 outer-membrane (OM) receptor responsible of selective transport to the periplasmic space.

MD simulations were performed from the structures determined by the authors experimentally so the work interdisciplinary and could be of interest to many audiences.

The simulation approach is appropriate and MD simulations are performed step-by-step with the aim of each clearly explained. The simulations are technically sound and extensive in terms of their length.

However, I have some comments regarding the systems and simulations

that have not been addressed or explained sufficiently in the manuscript.

- If the work is to reach a broader audience, it would be good to have a scheme showing the hypothesized transport route naming the receptors. Otherwise, it is difficult to follow the Introduction and beginning of the Results. The scheme of the Discussion is about the Results and it appears only at the end of the paper.

- It is not clear why only the POPE membrane was included and not a mix of lipids. The OM is not a homogenous POPE structure and CHARMM Gui can build membrane systems with different compositions. This is an approximation and should be discussed as limitations.

- The dynamics of the loops was found to be greatly affected by the LPS so omitting the outer core of OM is also another approximation which at least needs to be better discussed.

- Typically, calcium (divalent) ions are an important aspect in the simulations of membrane receptors. Here, these were not included and this also needs explanations.

- Following the above, I do not believe the authors may claim that LPS could promote opening of the BtuBG complex without performing simulations in a system that includes LPS and a membrane model reflecting the B. theta outer membrane. LPS was found to require calcium ions for neutralization and restrict the extracellular loop movement in other such systems. Therefore, one cannot assume that LPS

would promote opening of loops of such a large protein unless proven by MD simulations.

- line 92 - should be $+3e$, the unit is missing

- graphs/charts showing only one measure as a function of time or distance are not good quality and some are not necessary. For example, Figure 2: the distance in the top graph could go to SI and only the average with error can be given. The quality of the graphs showing the free energy profiles could be also improved and it would be good to add the structures of the conformational states pointing to certain areas of the graph.

- The average number of hydrogen bonds with error would be enough without the graph in Fig S3. In general Figure S3 is more informative in terms of B12-BtuG2 complex than Fig 2. Insets of Fig 2 - a, d, e, could be moved to SI or exchanged with better figures now placed in SI.

- It is not clear why MD simulations were performed in 300 K and not 310 K which would better reflect the 37C conditions of the bacteria.

- line 776 Why was the DG estimated at 300K and not 310K?

- In SMD it is necessary to show in SI the force profile as a function of the B12 position.

- how many steps of steepest descent method were used for minimization?

- How were the systems thermalized after minimization, this is either not reported or not clear.

Reviewer #2 (Remarks to the Author):

Sorry it took so long to respond but fell ill and finally catching up. The editor did a great job following up with me. My review will be short as the authors did a terrific job and is one of the better papers I have read in a while. Well written and very easy to read and understand. I have not looked at the biochemistry of vitamin B12-associated enzymes in >15 years so it brought back good memories. This is a wonderful paper describing the structure of a transporter and its surface lipoprotein cobalamin complexes from *Bacteroides* by cryoEM. The paper is thorough and shows insight into how the lipoprotein-mediated interactions facilitate the transfer of cobalamin. The paper is logical and should be published as is. (the pDB accession numbers are still needed but the proteomics experiments are straightforward I do not need to see the raw data myself)

Congrats on a wonderful paper.

Reviewer #3 (Remarks to the Author):

The manuscript by Abellon-Ruiz and colleagues is describing the structures and dynamics of the outer membrane B12 transporter BtuB1/2 in complex with the lipoprotein BtuG1/2, respectively. Although the structure of the B12 transporter BtuB from *E. coli* has previously been characterise, the two structures presented here from *B. thetaiotaomicron* describe a novel capture mechanism of B12 from the media by the use of a lipoprotein BtuG. This lipoprotein is not found in all gram-negative, presenting a new mechanism for B12 transport. BtuG is a lipoprotein associated with BtuB and has a very high affinity for B12. The authors present the crystal structure of BtuG2 with B12 that provides insights on its capture mechanism. They have also determined the crystal and cryo-EM structures of the lipoproteins with their associated transporter but in the absence of B12. In addition, they have determined the structure of BtuB1/G1 in complex with BtuH another accessory protein for B12 capture. Using molecular dynamics they provide a plausible mechanism for B12 transport from the capture proteins BtuG to the transporter BtuB.

This is a very exciting study and sheds light on a more complex process of B12 capture mechanism.

I believe this work should be published in Nature Communications upon revisions.

Major comments:

My main concern is the interpretation of the models in light of the biology and the lack of any functional data to address their conclusions for the BtuB1G1/BtuH complex. The B12-free complex structure shows BtuH on top of BtuG1, and based on their isolated structures with B12, they propose that B12 will be transported from BtuH to BtuG1 and then the transporter. In its current conformation B12 will not be able to transverse through the beta propeller of BtuG1. Is it possible the current conformation is an inhibitory mechanism? If BtuH is knockout what is the effect on B12 transport? Would mutations on that interface interrupt it or affect transport of B12? In my opinion this is exciting aspect of the work that the authors have missed out.

Additionally, they propose a series of interactions that can release the B12 from BtuG proteins to the transporter based on molecular dynamics. A series of mutations that could impact transport/cell growth for B12 would enhance their conclusions. These data are not validated and it would give a strong evidence on this mechanism.

The authors should provide a more quantified analysis of the interaction of BtuG2 with different corrinoids, eg ITC, SPR etc as it can provide more insights on the selectivity process including preference for a specific corrinoid.

The BtuB1G1 structure needs to be better refined. A clash score of 14.83 is unacceptable and a Ramachadran plot with only 86% in favoured areas is raising some questions on the model quality.

Minor comments:

Provide FoFc electron density maps for the B12 in complex with BtuG2

Supplementary Table 2 seems to be missing data for the BtuB1G1 in complex with BtuH

Provide a supplementary figure with the workflow of the cryo-EM processing including 2d class averages, micrographs, resolution FSC plots etc

The Electrostatic potential maps in Supplementary Figure 5 seem to have colours outside the molecules for most panels.

Reviewer #4 (Remarks to the Author):

The manuscript by Javier Abellon-Ruiz et al. " BtuB TonB-dependent transporters and BtuG surface lipoproteins form stable complexes for vitamin B12 uptake in gut Bacteroides." The authors present a study that reveals the structural basis for the interaction of VB12 molecules with lipoprotein BtuG, a protein from prominent human gut bacterium Bacteroides thetaiotaomicron localized to the outer leaflet of the outer membrane. The authors also analyzed the structures of the BtuB2-G2 and BtuB1-G1 complexes using X-ray crystallography and cryo-electron microscopy SPA, respectively, and the structures of both complexes showed stable closed conformations. Combining insights from another previously published TBDT system (SusCD) and molecular dynamics simulations presented in this work, the authors propose a lid-opening model of BtuB-G in which this system mediates the uptake of VB12. The gut microbiota has been an area of increasing interest in recent years, and therefore relevant structural biology studies have been valuable to the field to some extent. Furthermore, the acquisition of VB12 on the outer membrane of Gram-negative bacteria is an interesting mechanism.

The article is well-written and insightful. The structural aspects of the work are of high quality and the interpretations of the structures are impressive. The story is justified. However, I have some minor concerns that might help the article if the authors could address these issues before publication.

1. As mentioned by the authors in the article, an important question is how the substrate molecules enter the BtuBG complex. The authors propose a model for lid opening by incorporating insights previously gained from another TBDT system (SusCD) together with the molecular dynamics simulations of this work. Although no lid-opening conformation is seen by cryo-EM, it is still reasonable to assume that such a protein complex can be opened and accessible in certain spatiotemporal states. The molecular dynamics simulation work is remarkable, however, it is still relatively speculative. It is hard to rule out other possibilities, such as BtuG simply acting as a VB12-binding protein that triggers a conformational change upon binding to BtuB, so that VB12 is rapidly translocated upon entering the BtuBG complex. Also, the authors could consider an appropriate downgrading of the claimed model. However, if the authors can provide additional biochemical or EM evidence to support the model, it would be a nice option, too. A number of choices may be appropriate, such as performing 3D variability analysis in cryoSPARC, or even just by running an SDS-PAGE would be informative. Let's say, for instance, the BtuB2-G2 and BtuB1-G1 samples undergo trypsin digestion and run a gel, in the presence and absence of VB12 molecules, boiled and non-boiled conditions, etc.

2. Following the above topic, in lines 279, 314, the authors claimed that the adding of VB12 could lead to instability of the complex. Are there any other biochemical measurements that suggest or support this? And from Figure 4 c, the authors need to provide an example of the original cryo-EM image or so to

show that such poor 2D averages are indeed due to protein instability or dynamics and not because of data quality that is not comparable to the VB12-free samples.

3. From Figure 1b, the homologs of BtuH were not annotated at the locus. Are they in the neighborhood?

4. A representation of the electron density of the VB12 molecule would provide a sense for structural biologists and a more vivid picture for the binding of such ligands.

5. What is the sequence similarity between BtuG1 and BtuG2, especially those key amino acids involved in VB12 binding?

6. The structure of (PDB 3DSM) may be a ligand-free structure of BtuG2. Since a structure of BtuG in an apo state was not provided. Therefore, a structural comparison or superposition may be meaningful, for instance, BtuG (ligand-free), BtuG (VB12) and BtuG (B-G complex form).

7. The EM data panels, such as the representative cryo-EM micrograph, 2D class average pictures, particle direction distribution plot, FSC plot, local resolution map and representative EM densities vs models, were missing. EM processing flow-chart could be added in extended data figures or supplementary materials.

8. In line. 285 "A sarkosyl pre-extraction step did not improve the purity of the sample and suggests that the ~40 kDa band is an OMP." This would require a more general explanation. Otherwise, researchers from others field would wonder why the sarkosyl pre-extraction step can be an indicator of protein inner and outer membrane localization.

We would like to thank the reviewers for their constructive comments on our paper. In the following rebuttal document, our responses to the queries of the reviewers are in red font.

Reviewer #1 (Remarks to the Author):

The work of Abellon-Ruiz et al. focuses on the initial phase of the TonB-dependent transport of vitamin B12 in the *Bacteroides* bacteria of the gut. The authors use a variety of methods such as X-ray crystallography, cryo-EM, proteomics analyses and molecular dynamics (MD) simulations to determine the binding of vitamin B12 to the BtuG receptor and its transfer to BtuB.

The gut *B. theta* species have several vitamin B12 receptors in the outer-membrane, contrary to *E. coli*. In addition, they encode for lipoproteins that capture and bind the scarce vitamin B12 prior to delivering it to the outer-membrane receptor BtuB. The authors focused on the structure, interactions and vitamin B12 transfer between the BtuG2 lipoprotein scavenging B12 and the BtuB2 outer-membrane (OM) receptor responsible of selective transport to the periplasmic space.

MD simulations were performed from the structures determined by the authors experimentally so the work interdisciplinary and could be of interest to many audiences.

The simulation approach is appropriate and MD simulations are performed step-by-step with the aim of each clearly explained. The simulations are technically sound and extensive in terms of their length.

However, I have some comments regarding the systems and simulations that have not been addressed or explained sufficiently in the manuscript.

We thank the reviewer for the insightful comments, which helped to improve the manuscript significantly.

If the work is to reach a broader audience, it would be good to have a scheme showing the hypothesized transport route naming the receptors. Otherwise, it is difficult to follow the Introduction and beginning of the Results. The scheme of the Discussion is about the Results and it appears only at the end of the paper.

We are not sure what the reviewer means here. The transport mechanism is one of the main findings of the paper, and showing this in the introduction would be confusing (we tried). We have written the paper as one usually does, *i.e.* with the introduction ending with a summary paragraph that should help the reader decide whether to continue reading. The scheme is in our opinion where it should be, *i.e.* in the discussion.

Query 2: It is not clear why only the POPE membrane was included and not a mix of lipids. The OM is not a homogenous POPE structure and CHARMM Gui can build membrane systems with different compositions. This is an approximation and should be discussed as limitations.

In principle we agree with the reviewer. However, the main problem is that the composition of the *B. theta* OM is far from clear. The *Bacteroidetes* are phylogenetically far removed from *Proteobacteria*, and it should not be assumed that the OMs of, *e.g.*, *E. coli* and *B. theta* are similar. Even though the *B. theta* OM is likely to be asymmetric and contains highly negatively charged lipooligosaccharide (LOS) molecules

in the outer leaflet (Pither et al., 2022; new reference 30), there is no data on the extent of charge neutralisation via divalent metal ions as occurring in Proteobacteria. Moreover, *B. theta*, like other *Bacteroides*, is known to contain abundant sphingolipids (An et al., 2011; Brown et al., 2019; Ryan et al., 2023; new refs. 65-67), and there is no reason to assume that these won't be present in the OM. Thus, while using a POPE bilayer for the *B. theta* OM is undoubtedly inaccurate, using an *E. coli*-like asymmetric OM model with rough LPS is likely to be non-physiological as well. In addition, our main goal was to characterize the B₁₂ acquisition process and the translocation to BtuB. The inclusion of LPS/LOS molecules in the OM model would have led to additional convergence issues due to the known, very slow diffusion of these molecules in the membrane. Therefore, we think that using a symmetrical POPE bilayer is an acceptable compromise. This is now mentioned in the text (lines 400-402) and described in more detail in the methods (page 35).

The dynamics of the loops was found to be greatly affected by the LPS so omitting the outer core of OM is also another approximation which at least needs to be better discussed.

Again, we agree with the reviewer, but refer to our response to the previous point.

Typically, calcium (divalent) ions are an important aspect in the simulations of membrane receptors. Here, these were not included and this also needs explanations.

We assume that the reviewer refers to the charge neutralisation of LPS by divalent metal ions, and not to the very specific role of calcium ions in the binding of B₁₂ by *E. coli* BtuB. As stated above, there is no data on the extent of charge neutralisation of *B. theta* LOS by divalent metal ions.

Following the above, I do not believe the authors may claim that LPS could promote opening of the BtuBG complex without performing simulations in a system that includes LPS and a membrane model reflecting the *B. theta* outer membrane. LPS was found to require calcium ions for neutralization and restrict the extracellular loop movement in other such systems. Therefore, one cannot assume that LPS would promote opening of loops of such a large protein unless proven by MD simulations.

The original statement was speculative and this has now been made clearer (line 402 and lines 590-2). As stated above, we note that there is no reliable model for the *B. theta* OM. We do not think that this warrants any further simulations, also taking into account the large amount of data already in our paper and the convergence issues of using LPS-containing membrane models.

line 92 - should be +3e, the unit is missing.

This has been corrected.

graphs/charts showing only one measure as a function of time or distance are not good quality and some are not necessary. For example, Figure 2: the distance in the top graph could go to SI and only the average with error can be given. The quality of the graphs showing the free energy profiles could be also improved and it would be good to add the structures of the conformational states pointing to certain areas of the graph.

Figure 2d has been updated as suggested by the reviewer.

The average number of hydrogen bonds with error would be enough without the graph in Fig S3. In general Figure S3 is more informative in terms of B12-BtuG2 complex than Fig 2. Insets of Fig 2 - a, d, e, could be moved to SI or exchanged with better figures now placed in SI.

We have made changes to Figure 2 and Sup. Fig. 3 to hopefully make things clearer. One panel (d) has been added to Sup. Fig. 3.

It is not clear why MD simulations were performed in 300 K and not 310 K which would better reflect the 37C conditions of the bacteria.

The reviewer is correct, and we apologise for the mistake on our part. However, we expect that no major changes will be observed in the results for the simulations at 300 K and at 310 K due to the small change in the temperature. For example, to examine BtuG2 lid opening in the BtuB2G2 complex, we performed unbiased simulations at 300 K, 350 K, and 400 K. No clear differences were observed for the simulations at 300 K and at 350 K (Supplementary Fig. 9d).

Line 776: Why was the DG estimated at 300K and not 310K?

The dissociation constant was calculated at 300 K as the WTMD simulations were performed at 300 K, please see the previous point.

In SMD it is necessary to show in SI the force profile as a function of the B12 position.

The SMD results have been removed from the paper. We have now much better (experimental) data for B₁₂ binding by BtuBG and transfer of B₁₂ from BtuG to BtuB (Figs. 6 and 7).

How many steps of steepest descent method were used for minimization?

50000 steps for the steepest descent energy minimization were used, while the systems were well minimized before that. The force F_{\max} and the potential energy of the systems were monitored to examine the minimization process. (page 36)

How were the systems thermalized after minimization, this is either not reported or not clear.

Before starting the production MD runs, all systems were well equilibrated for a total of 60 ns at 300 K. In more detail, after minimization, the systems consisting of protein, lipid, ions, and water were equilibrated by a two-step constant volume (NVT) equilibration of 5 ns each with a time step of 1 fs, where velocity rescaling was used to maintain the temperature at 300 K and the calculated average system temperature was 300 K. After that, a four-step constant pressure (NPT) equilibration of 50 ns was performed by removing the restraints on the protein, the membrane, and the ligands in a stepwise manner. Thus, the systems were thermalized in one step in the present study while care was taken that no significant artifacts appeared by gradually releasing the constraints (page 36).

Reviewer #2 (Remarks to the Author):

Sorry it took so long to respond but fell ill and finally catching up. The editor did a great job following up with me. My review will be short as the authors did a terrific job and is one of the better papers I have read in a while. Well written and very easy to read and understand. I have not looked at the biochemistry of vitamin B12-associated enzymes in >15 years so it brought back good memories. This is a wonderful paper describing the structure of a transporter and its surface lipoprotein cobalamin complexes from *Bacteroides* by cryoEM. The paper is thorough and shows insight into how the lipoprotein-mediated interactions facilitate the transfer of cobalamin. The paper is logical and should be published as is. (the pride accession numbers are still needed but the proteomics experiments are straightforward I do not need to see the raw data myself)

Congrats on a wonderful paper.

We thank the reviewer for the positive comments.

Reviewer #3 (Remarks to the Author):

The manuscript by Abellon-Ruiz and colleagues is describing the structures and dynamics of the outer membrane B12 transporter BtuB1/2 in complex with the lipoprotein BtuG1/2, respectively. Although the structure of the B12 transporter BtuB from *E. coli* has previously been characterise, the two structures presented here from *B. thetaiotaomicron* describe a novel capture mechanism of B12 from the media by the use of a lipoprotein BtuG. This lipoprotein is not found in all gram-negative, presenting a new mechanism for B12 transport. BtuG is a lipoprotein associated with BtuB and has a very high affinity for B12. The authors present the crystal structure of BtuG2 with B12 that provides insights on its capture mechanism. They have also determined the crystal and cryo-EM structures of the lipoproteins with their associated transporter but in the absence of B12. In addition, they have determined the structure of BtuB1/G1 in complex with BtuH another accessory protein for B12 capture. Using molecular dynamics they provide a plausible mechanism for B12 transport from the capture proteins BtuG to the transporter BtuB.

This is a very exciting study and sheds light on a more complex process of B12 capture mechanism.

We thank the reviewer for the positive comments.

I believe this work should be published in Nature Communications upon revisions.

Major comments:

My main concern is the interpretation of the models in light of the biology and the lack of any functional data to address their conclusions for the BtuB1G1/BtuH complex. Their B12-free complex structure shows BtuH on top of BtuG1, and based on their isolated structures with B12, they propose that B12 will be transported from BtuH to BtuG1 and then the transporter. In its current conformation B12 will not be able to transverse through the beta propeller of BtuG1. Is it possible the current conformation is an inhibitory mechanism?

This is a possibility but we're not sure why there would be an inhibitory mechanism, given that *B. theta* devotes a lot of cellular resources to acquire B₁₂ (Fig. 1b). We consider it more likely that the docked conformation represents, *e.g.*, a resting state of the transporter. It is also possible that this particular conformation has been selected for ("frozen out") during the process of cryoEM grid preparation. Please note that the BtuH is part of BtuG (it is one chain and not a complex, see the next point).

If BtuH is knockout what is the effect on B12 transport?

Please note that there are three BtuH proteins/domains in *B. theta*: one each in locus 1 and 2, plus the one in locus 1 where the BtuH is fused to BtuG (stated in the text and indicated in Fig. 1b via colouring *btuH* sequences pink), and for which we have determined the BtuB1G1 cryoEM structure of Fig. 4. Thus, this is a complex question to answer. However, BtuH from locus 2 has been deleted, and this results in growth defects *in vitro* and fitness defects *in vivo* (ref. 26). There's no data on either the BtuH protein/domain from locus 1, but given that deletion of the entire locus 1 (in the wild-type background) does not result in a phenotype (ref. 18), we expect that the same will apply to BtuH1 deletions.

Would mutations on that interface interrupt it or affect transport of B12? In my opinion this is exciting aspect of the work that the authors have missed out.

This is indeed an interesting question that could be explored in a locus2,3 deletion background in which BtuH1 has been deleted as well (leaving only BtuG1 with its H-domain). The role of BtuH and other surface lipoproteins in B₁₂ acquisition will be investigated in future work. Given the dominant role of locus 2 in B₁₂ acquisition (page 23 and refs. 18, 19), we intend to focus mostly on BtuH2.

Additionally, they propose a series of interactions that can release the B12 from BtuG proteins to the transporter based on molecular dynamics. A series of mutations that could impact transport/cell growth for B12 would enhance their conclusions. These data are not validated and it would give a strong evidence on this mechanism.

The new cryo-EM structure for CNCbl-bound BtuB3G3, combined with unbiased MD simulations included in the revision answers this query (new Figs. 6 and 7). The data show that the EL8 loop of BtuB3 is crucial to release the vitamin from its binding pocket in BtuG3 (where it is bound with extremely high affinity), and we propose that this mechanism operates in the other BtuBG transporters as well. This data has been validated by functional data as suggested by the reviewer (Fig. 6 f,g).

The authors should provide a more quantified analysis of the interaction of BtuG2 with different corrinoids, eg ITC, SPR etc as it can provide more insights on the selectivity process including preference for a specific corrinoid.

This has already been done. In Wexler et al. (ref. 19) Kd values for the BtuG2-cobinamide and BtuG2-cyanocobalamin interactions were measured with SPR to be sub-pM (1.93×10^{-13} M and 1.87×10^{-13} M respectively). Note that these affinities are challenging to measure with SPR and too high for ITC. In any case, these data show that different corrinoids bind with similar, very high affinities to BtuG with likely little to no selectivity, which is supported by our crystal structures (Supplementary Fig. 1).

The BtuB1G1 structure needs to be better refined. A clash score of 14.83 is unacceptable and a Ramachandran plot with only 86% in favoured areas is raising some questions on the model quality.

We have improved the model as suggested by the reviewer. Clash score of the improved model is 7.0 and the Ramachandran favoured region has now 90.7 % of residues. Please note that part of the structure (especially the H domain) is around 3.5-4.5 Å resolution (Supplementary Fig. 7), so we don't think that the original stats were that bad.

Minor comments:

Provide FoFc electron density maps for the B12 in complex with BtuG2

This (and a 2FoFc map) has been added to Supplementary Fig. 1.

Query 6: Supplementary Table 2 seems to be missing data for the BtuB1G1 in complex with BtuH

As commented above, BtuG1 has two domains, BtuG and BtuH (it is not a complex but a fusion).

Query 7: Provide a supplementary figure with the workflow of the cryo-EM processing including 2d class averages, micrographs, resolution FSC plots etc

We have added three new supplementary figures with this information, one for BtuB1G1 (Sup. Fig. 7) and two for the new structure of BtuB3G3-CNCbl (Sup. Figs. 12 and 13).

Query 8: The Electrostatic potential maps in Supplementary Figure 5 seem to have colours outside the molecules for most panels.

That is correct. These figures show the electrostatic potentials created by the charges present in the vitamin and the protein. In panel a, this potential is projected onto the molecular structure but in principle the electric potential, whose negative gradient is the electric field, is present in the space around the B₁₂ ligand and protein. For example, a single charge creates an electrostatic potential which decays like $1/r$, with r being the distance from that charged particle.

Reviewer #4 (Remarks to the Author):

The manuscript by Javier Abellon-Ruiz et al. " BtuB TonB-dependent transporters and BtuG surface lipoproteins form stable complexes for vitamin B12 uptake in gut Bacteroides." The authors present a study that reveals the structural basis for the interaction of VB12 molecules with lipoprotein BtuG, a protein from prominent human gut bacterium Bacteroides thetaiotaomicron localized to the outer leaflet of the outer membrane. The authors also analyzed the structures of the BtuB2-G2 and BtuB1-G1 complexes using X-ray crystallography and cryo-electron microscopy SPA, respectively, and the structures of both complexes showed stable closed conformations. Combining insights from another previously published TBDT system (SusCD) and molecular dynamics simulations presented in this work, the authors propose a lid-opening model of BtuB-G in which this system mediates the uptake of VB12. The gut microbiota has been an area of increasing interest in recent years, and therefore relevant structural biology studies have been valuable to the field to some extent. Furthermore, the acquisition of VB12 on the outer membrane of Gram-negative bacteria is an interesting mechanism.

The article is well-written and insightful. The structural aspects of the work are of high quality and the interpretations of the structures are impressive. The story is justified. However, I have some minor concerns that might help the article if the authors could address these issues before publication.

As mentioned by the authors in the article, an important question is how the substrate molecules enter the BtuBG complex. The authors propose a model for lid opening by incorporating insights previously gained from another TBDT system (SusCD) together with the molecular dynamics simulations of this work. Although no lid-opening conformation is seen by cryo-EM, it is still reasonable to assume that such a protein complex can be opened and accessible in certain spatiotemporal states. The molecular dynamics simulation work is remarkable, however, it is still relatively speculative. It is hard to rule out other possibilities, such as BtuG simply acting as a VB12-binding protein that triggers a conformational change upon binding to BtuB, so that VB12 is rapidly translocated upon entering the BtuBG complex. Also, the authors could consider an appropriate downgrading of the claimed model.

We're not quite sure what the reviewer means here. BtuG is most definitely a B₁₂-binding protein. However, our data show that BtuG forms a very stable complex with BtuB (Fig. 3a). Thus, BtuB is always bound to BtuG, and the only way for a BtuBG complex to bind B₁₂ is for a lid-opening of the complex, analogous to SusCD-like systems. What the reviewer appears to propose for BtuG we propose for BtuH instead (lines 578-9), because we know that BtuH is not stably associated with BtuBG. However, we do not know whether there is an excess of BtuG relative to BtuB (the BtuBG complexes are tagged on BtuB), so we cannot exclude a potential role for any excess, "stand-alone" BtuG in BtuBG opening, but consider this possibility less likely.

However, if the authors can provide additional biochemical or EM evidence to support the model, it would be a nice option, too. A number of choices may be appropriate, such as performing 3D variability analysis in cryoSPARC, or even just by running an SDS-PAGE would be informative. Let's say, for instance, the BtuB2-G2 and BtuB1-G1 samples undergo trypsin digestion and run a gel, in the presence and absence of VB12 molecules, boiled and non-boiled conditions, etc.

Our new cryoEM structure for the CNCbl-bound BtuB3G3 complex shows that this complex can be loaded by using a mild detergent, which means that it opens without any external factors, at least *in vitro*. Importantly, the comparison of this structure with that of apo BtuB2G2 (both structures are very similar) allows us to propose a mechanism for substrate acquisition by BtuG and transfer from BtuG to BtuB (new Figs. 6 and 7). We corroborate the proposed mechanism with functional assays and via MD simulations.

This leaves the mechanism of actual opening of BtuBG complexes as the one aspect for which we do not have data. We think that, for the current study, we have to accept that BtuBG complexes need to open to bind substrate, but that we do not yet know how the complexes open and are loaded with B₁₂ *in vivo*. We offer some possibilities in the discussion (lines 573-9), but investigating these will require substantial work that is beyond the scope of the current study.

Following the above topic, in lines 279, 314, the authors claimed that the adding of VB12 could lead to instability of the complex. Are there any other biochemical measurements that suggest or support this? And from Figure 4 c, the authors need to provide an example of the original cryo-EM image or so to show that such poor 2D averages are indeed due to protein instability or dynamics and not because of data quality that is not comparable to the VB12-free samples.

We have added a panel showing (via SEC) that the BtuB3G3 complex, which has the highest expression levels and appears qualitatively more stable than BtuB1G1 and BtuB2G2, dissociates in the presence of LDAO and B₁₂ (Sup. Fig. 11a). The BtuB3G3 complex can be loaded with CNCbl by using a mild detergent.

The text has been amended (lines 327-9) to state that the quality of the data for BtuB1G1-B₁₂ could indeed have resulted in poorer 2D class averages compared to the sample without B₁₂. However, with the data now included for BtuB3G3 in the presence of LDAO and B₁₂, it seems reasonable to assume that the lower quality of the BtuB1G1-B₁₂ data could indeed be due to the presence of B₁₂. In any case, this point seems moot given our new cryoEM structure for BtuB3G3-B₁₂.

From Figure 1b, the homologs of BtuH were not annotated at the locus. Are they in the neighborhood?

They have been annotated and are now coloured in pink.

A representation of the electron density of the VB12 molecule would provide a sense for structural biologists and a more vivid picture for the binding of such ligands.

This has been added to Supplementary Fig. 1.

What is the sequence similarity between BtuG1 and BtuG2, especially those key amino acids involved in VB12 binding?

With the new cryoEM data for BtuB3G3-CNCbl, we have focused on the more relevant, high similarity between BtuG2 and BtuG3 (page 17-18). The homology between BtuG1 and BtuG2 is 40%, but the residues involved in binding are reasonably conserved. We think that with the new data and the shift in emphasis to the BtuBG systems from locus 2 and 3, adding this information would not improve the paper and would seem more suitable for an in depth study of BtuB1G1 on its own. Please see below for an

alignment of the three BtuG paralogs. A similar alignment for just BtuG2 and BtuG3 is now shown in Supplementary Fig. 11.

The structure of (PDB 3DSM) may be a ligand-free structure of BtuG2. Since a structure of BtuG in an apo state was not provided. Therefore, a structural comparison or superposition may be meaningful, for instance, BtuG (ligand-free), BtuG (VB12) and BtuG (B-G complex form).

We have added this to the supplementary material (Supplementary Fig. 1f). Both structures are identical. We have now also included comparisons with BtuG3 (Supplementary Fig. 11). Our data suggest that BtuG2 and BtuG3 bind B₁₂ in identical fashion, which is also the same in the presence and absence of the cognate BtuB.

The EM data panels, such as the representative cryo-EM micrograph, 2D class average pictures, particle direction distribution plot, FSC plot, local resolution map and representative EM densities vs models, were missing. EM processing flow-chart could be added in extended data figures or supplementary materials.

We apologise for the oversight. As requested, these have been added for the BtuB1G1 and BtuB3G3-B₁₂ cryo-EM structures (Supplementary Figs. 7, 12, 13).

In line 285 “A sarkosyl pre-extraction step did not improve the purity of the sample and suggests that the ~40 kDa band is an OMP.” This would require a more general explanation. Otherwise, researchers from others field would wonder why the sarkosyl pre-extraction step can be an indicator of protein inner and outer membrane localization.

This has now been clarified in the Figure 3 legend and a reference has been added (ref 24).

REVIEWERS' COMMENTS

Reviewer #1 (Remarks to the Author):

The authors addressed my comments and followed the suggestions. They included the changes in the manuscript so now I can recommend publication.

Reviewer #3 (Remarks to the Author):

The authors have addressed all my queries. The inclusion of the additional biochemical data and structural work, BtuB3G3 in complex with CNcbl, has further enhanced this excellent manuscript. Well done to the authors for this excellent work.

Reviewer #4 (Remarks to the Author):

So basically the revised version is quite impressive for this reviewer. In particular, I would not have expected the authors to provide new and important cryoelectron microscopy as well as crystal structures. The quality of these structures is excellent.

I have no more questions to mention and my previous concerns are clearly addressed, especially after seeing the newly added high resolution structure of BtuB3G3-CNCbl complex. This manuscript should be published with no further delay.

In short, this manuscript is now dramatically improved. I must congratulate the authors in advance for their efforts and exquisite work in understanding how vitamin B12 molecules can be transported across OM.